# DISTRIBUTED IN-CONTEXT LEARNING UNDER NON-IID AMONG CLIENTS

## ABSTRACT

Advancements in large language models (LLMs) have shown their effectiveness in multiple complicated natural language reasoning tasks. A key challenge remains in adapting these models efficiently to new or unfamiliar tasks. In-context learning (ICL) provides a promising solution for few-shot adaptation by retrieving a set of data points relevant to a query, called in-context examples (ICE), from a training dataset and providing them during the inference as context. Most existing studies utilize a centralized training dataset, yet many real-world datasets may be distributed among multiple clients, and remote data retrieval can be associated with costs. Especially when the client data are non-identical independent distributions (non-IID), retrieving from clients a proper set of ICEs needed for a test query presents critical challenges. In this paper, we first show that in this challenging setting, test queries will have different preferences among clients because of non-IIDness, and equal contribution often leads to suboptimal performance. We then introduce a novel approach to tackle the distributed non-IID ICL problem when a data usage budget is present. The principle is that each client's proper contribution (budget) should be designed according to the preference of each query for that client. Our approach uses a data-driven manner to allocate a budget for each client, tailored to each test query. Through extensive empirical studies on diverse datasets, our framework demonstrates superior performance relative to competing baselines.

## 1 INTRODUCTION

Recent significant progress in large language models (LLMs) (Achiam et al., 2023; Touvron et al., 2023a;b; Team et al., 2023) has demonstrated their effectiveness across various natural language processing (NLP) tasks (Wang et al., 2018; 2019). Despite their impressive performances, they still require adaptation to the specific downstream tasks for better performance. However, adaptation poses challenges due to LLMs' vast number of trainable parameters.

In-context learning (ICL) (Dong et al., 2022) is a notable method that distinguishes itself through both its effectiveness and efficiency. In brief, ICL adapts to the target task by incorporating context information following two primary steps: i) identify samples from the training dataset helpful to solve the target query by creating a prompt describing a context; ii) feed the constructed prompt with the target query and get the answer. Previous related works on ICL mainly have focused on the construction of a prompt describing the context, which involves several sub-problems, such as the retrieval of in-context examples (ICEs) (Robertson et al., 2009) and determining the optimal sequence for the selected ICEs (Zhang et al., 2024).

A common assumption in most existing ICL research is that the system has access to a high-quality centralized dataset used for retrieval. However, in many application scenarios, such as health informatics, centralized datasets may not be feasible, and data could be distributed in different institutions, which calls for the distributed ICL. In addition, when the data is proprietary and possesses high value towards inferences, access to data entries may also be bound to data pricing strategies (Xu et al., 2023; Cong et al., 2022). For instance, the system needs to pay the local institution based on the number of samples sent to the system to share profits from inferences (Tang et al., 2020). Under this scenario, aggregating ICEs from local clients to a center server for ICL entails significant financial costs and lacks efficiency.

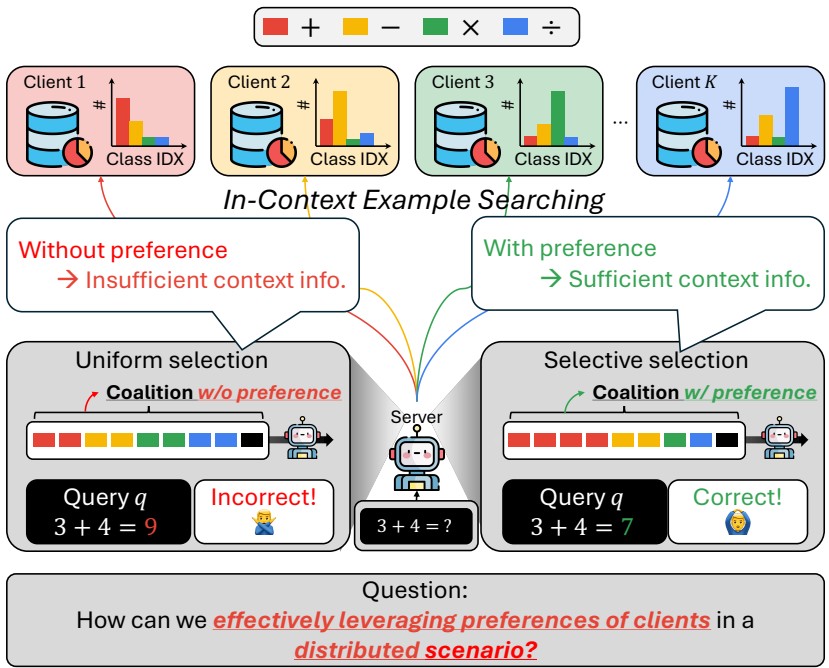

Figure 1: Problem overview. When datasets are distributed among clients in a non-IID manner, it creates an obstacle in generating a good context (left). However, by assigning appropriate budgets to leverage per-client expertise, better context can be created (right).

In this paper, we focus on integrating knowledge from distributed clients to achieve better ICL performance under the per-query ICE budget constraint. Specifically, we formalize the distributed ICL problem where the ICEs are distributed on clients, and the server has an LLM for ICL inference but can only request a limited number of ICEs from all clients for each query, which we refer to as the *ICE budget*.

We begin by identifying the key challenge in distributed ICL with ICE budget constraints lies in the non-independently and identically distributed (non-IID) training data, as shown in Section 3.1. For example, in Figure 1, data samples are spread across $C$ clients, each with a unique data distribution. Specifically, client 1 primarily contains $(+)$ samples, while client 2 is mainly constituted by $(-)$ examples. Only limited research (Mohtashami et al., 2023) tried to address the challenge of distributed datasets for ICL, while none considers the challenging real-world setting of non-IID clients. This leaves a critical question unanswered: *What happens to distributed ICL when local clients are non-IID?*

To further the understanding of the key challenge in **the distributed non-IID ICL**, we explore the local retrieval process on non-IID clients. We found that each query has different preferences for different clients based on local knowledge distribution, that is, the number of samples needed from different clients should vary based on local sample distribution. As the toy example shown in Figure 1, when the server creates context by uniformly assigning budgets to clients, the answer might be incorrect due to the insufficiency of $(+)$ information in the context. To be more detailed, the server assigns the clients who have expertise on $(-)$, $(\times)$, and $(\div)$ operations with the same budget as on $(+)$, without any preference. Nevertheless, if the server assigns more budget to clients with many $(+)$ samples, such as client 1, it can create a more relevant context to answer the query related to $(+)$ operation. This indicates that under non-IID, the server should allocate the budgets over clients based on the preference of each query itself, as well as the distribution of local training samples.

Motivated by this, we propose a novel distributed ICL framework to collaboratively collect scattered information among non-IID clients by properly assigning ICE budgets to each client. First, the server will gather the optimal budget statistics using an existing proxy dataset on the server side. Next, the server will use this dataset to train the budget allocator. During the deployment stage, the server will

predict the proper budget for each client using this trained budget allocator given each test query and perform ICL among clients. Furthermore, in practical scenarios where privacy concerns arise, we augment our framework with the paraphrasing method (Mohtashami et al., 2023) to secure privacy.

**Contributions.** We summarize our contributions as follows.

- To the best of our knowledge, we are the first to study the challenging real-world setting of ICL with distributed non-IID clients. We identify the principal challenge as properly assigning the ICE budget for non-IID clients based on the preference of each test query and local knowledge distribution.

- We propose a framework to handle the distributed non-IID ICL. This framework trains a budget allocator on the server with the help of a server-side proxy dataset. Then, the server will use this trained allocator to decide how many ICEs to retrieve from each client for the ICL process, enabling collaborative action among clients.

- Across a range of dataset benchmarks featuring various non-IID configurations as well as on different LLM architectures, our approach has been validated to enhance ICL performance. Notably, we examine both non-private, *i.e.,* communicate raw samples directly, and private cases using the paraphrasing method to secure privacy. In both scenarios, our approach shows superiority to the previous method and other reasonable baselines.

## 2 PROBLEM FORMULATION

In this section, we provide a detailed problem formulation. First, we begin with the specifics of in-context learning (ICL), followed by a description of distributed non-IID ICL.

### 2.1 IN-CONTEXT LEARNING

**Notation.** We consider a NLP tasks which have training dataset $\mathcal{D} = \{(x_i, y_i)\}_{i=1}^{N}$ with $N$ training samples. Here, $x_i$ is the input text, and $y_i$ is the corresponding output. In the test phase, a test query $x_q$ is given.

**Retrieval.** We employ the off-the-shelf pre-trained retriever KATE (Liu et al., 2021)[1], which utilizes $k$-NN example selection. This retriever employs a sentence encoder $\mathcal{E}(\cdot)$ to measure the similarity between the in-context example $x_i$ in dataset $\mathcal{D}$ and the query $x_q$ as follows:

$$d(e_i, e_q) = \|e_q - e_i\|_2, \tag{1}$$

where $e_q = \mathcal{E}(x_q)$ and $e_i = \mathcal{E}(x_i)$. We select $k$ samples using the following criterion:

$$\mathcal{T}(e_q, k|\mathcal{D}) = \underset{e_i = \mathcal{E}(x_i) \forall (x_i, y_i) \in \mathcal{D}}{\arg \operatorname{Top-}k}(d(e_i, e_q)), \tag{2}$$

where $\mathcal{T}(e_q, k|\mathcal{D})$ denotes the selected samples from the dataset $\mathcal{D}$, and used for inference.

**ICL Inference.** In the test phase, given a test query with input $x_i$, relevant $k$ training samples called in-context examples (ICEs) are selected, *i.e.,* $S = \mathcal{T}(e_q, k|\mathcal{D})$. Based on the retrieved samples, we feed the constructed context prompt $s(S, x_q)$ into LLM for inference and obtain results via:

$$y_t = \arg \max_y p_{\text{LLM}}(y|s(S, x_q), y_{<t}) \quad \text{where} \quad s(S, x_q) = (x_1, y_1) \odot \ldots \odot (x_k, y_k) \odot x_q, \tag{3}$$

where the $\odot$ operation denotes concatenation, and $s(S, x_q)$ is the context constructed using query $x_q$ and samples in $S$; the term $p_{\text{LLM}}$ represents the output softmax probability of the LLM, functioning autoregressive, meaning that the output up to time $t$, *i.e.,* $y_{<t}$, is input back into the model to generate the $t^{\text{th}}$ output, $y_t$. Previous works (Ye et al., 2023; Levy et al., 2022) on ICL mainly focus on the selection of $S$ under a centralized setting. However, we investigate the scenario where $\mathcal{D}$ is split among several clients, each following non-IID distributions.

---

[1]We do not fine-tune the retriever for each task, which is impractical because we cannot gather the distributed datasets.

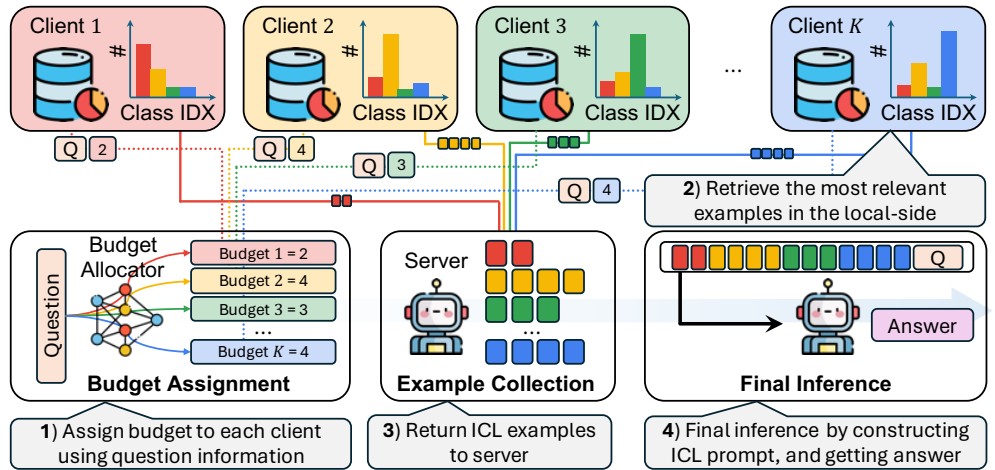

Figure 2: Overview of the pipeline: First, the **budget allocator** assigns a budget to each client based on the question. Subsequently, each client retrieves their relevant samples and sends them back to the server. The server infers the answer by feeding the question, which is composed of concatenated context examples and the query.

## 2.2 DISTRIBUTED NON-IID ICL

**Distributed ICL Setting.** We consider $C$ clients with a centralized server in our system. Each client $c \in [C]$ has local training dataset $\mathcal{D}_c = \{(x_i^c, y_i^c)\}_{i=1}^{N_c}$ with $N_c$ training samples. Note that $\mathcal{D}_c$ follows different distributions for different clients. We follow the non-IID conditions as defined in Li et al. (2022), with details provided in Appendix A. In summary, we allocate data on a per-class basis, where each client receives a specific number of classes, meaning each client has samples from only specified classes. Clients and the server have identical off-the-shelf pre-trained retrievers. Consider the computation resource limitation on clients as in many real scenarios (Yoo et al., 2022), only the server is equipped with an LLM. Moreover, the server has limited proxy dataset $\mathcal{D}_{\text{proxy}} = \{(x_j^{\text{proxy}}, y_j^{\text{proxy}})\}_{j=1}^{N_{\text{proxy}}}$, that $N_{\text{proxy}} \ll \sum_{c=1}^{C} N_c$. The server has quite a small $\mathcal{D}_{\text{proxy}}$, and it is an auxiliary dataset to extract information for collaboration to make the problem feasible.

**Pipeline.** First, the server requests relevant samples from each client by sending $x_q$ to all clients with local budgets $k_c$. Remark that each query $x_q$ has its own preference of each client, which can be represented as $k_c$. A larger $k_c$ indicates the given test query $x_q$ prefers more information from client $c$, compared with client $c'$ with a smaller $k_{c'}$. Here, $x_q$ can be anonymized by paraphrasing, as done in previous works (Mohtashami et al., 2023)[2]. Each client then selects the most relevant $k_c$ samples from their local training dataset, *i.e.*, $S_c = \mathcal{T}(e_q, k_c | \mathcal{D}_c) \subset \mathcal{D}_c$, and returns them to the server. The server receives $S_c$ from clients and generates the context $s$ based on the merged examples, $S = \bigcup_{c=1}^{C} S_c$. In the final step, the server infers $y$ using $s(S, x_q)$. The entire framework also can be described in Figure 2. In this paper, we are concentrating on assigning $k_c$ to each client as described in Figure 2.

## 3 OBSERVATIONS

In this section, we describe several empirical supports to handle the distributed non-IID ICL. First, we demonstrate that non-IID distributions hinder the merging of scattered information. We then establish our goal, termed as ***oracle budget***, which reflects the server's preference for each client if the server knows all distributed data. Finally, we check if predicting the oracle budget of each test query for inference is feasible.

---

[2]Although our main experiments utilize the non-paraphrased dataset, we also present the paraphrased results in Section 5.

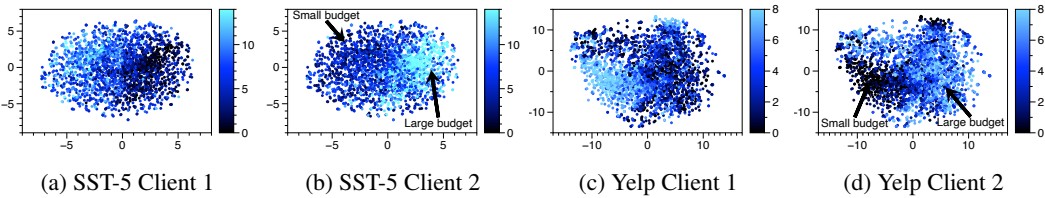

Figure 3: Non-IID experimental results. It shows that centralized performance is comparable to the IID case, whereas non-IID scenarios exhibit a significant declined performance. This highlights the critical importance of addressing non-IIDness to find a solution.

### 3.1 NON-IIDNESS LEADS TO PERFORMANCE DROP

First of all, we evaluate the effect of non-IIDness. Straightforwardly, we distribute the budget $\{k_c\}_{c=1}^C$ uniformly according to the following criteria: Given $C$ clients are involved in answering this question, and the number of samples for context is $k$. We first explore the naïve equally assigned local budget scheme in both IID and non-IID settings. That is, each client $c \in [C]$ locally retrieves top-$k_c$ samples where $k_c = \lceil \frac{k}{C} \rceil$ from local dataset $\mathcal{D}_c$. Detailed experimental settings are described in Appendix B.

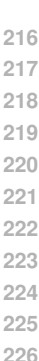

| (a) SST-5 Client 1 | (b) SST-5 Client 2 | (c) Yelp Client 1 | (d) Yelp Client 2 |

Figure 4: t-SNE analysis of each client across two datasets. Each figure demonstrates that the budgets can be segregated by training a simple classifier, as they exhibit clustered subgroup pattern.

As illustrated in Figure 3, we observe the followings: (1) There is no significant performance degradation between the centralized case ( ) and the IID case ( ). This is expected, as the merged top-$k_c$ samples in the IID case closely resemble the centralized top-$k$ samples. Any minor discrepancies are attributed to differences in sample ordering. (2) However, performance degradation becomes pronounced in non-IIDness case (refer to the comparison between , and ). Hereinafter, we gather insights to address the distributed non-IID ICL.

### 3.2 PROPER BUDGET PER QUERY FOR EACH CLIENT

**Oracle budget.** The remaining issue is that to make the server operate similar with the centralized manner, it needs to allocate the budget as if it knows complete knowledge of all clients. We call this budget for each client as the *oracle budget* for query embedding $e_q$ and define it as follows:

$$k_c^\star(e_q) = \left| \mathcal{T}(e_q, k|\mathcal{D}_c) \cap \mathcal{T}(e_q, k|\mathcal{D}) \right|,$$

where $\mathcal{T}(\cdot)$ is defined as Eq. (2) and $|\cdot|$ is set cardinality. Note that the physical meaning of $k_c^\star(e_q)$ is the number of shared samples between the top-$k$ relevant to $e_q$ in local $\mathcal{D}_c$ and global $\mathcal{D}$ datasets.

**Check of predictability of oracle budget.** For the next step, it is necessary to check if $e_q$ has sufficient patterns of oracle budget to extract and use it in the inference phase. Our hypothesis is that similar queries may share similar oracle budget patterns and preferences on the same client, and it can lead to similar budget allocations for that client. Therefore, to verify this hypothesis, we perform t-SNE analysis (Van der Maaten & Hinton, 2008) on the embeddings obtained from the retriever for queries. Furthermore, we color each sample based on the oracle budget $k_c^\star(e_q)$. As described in Figure 4, similar query embeddings exhibit similar oracle budget patterns. This indicates that, given

a test query, we can infer the budget assignment for each client. However, it is challenging to predict fine-grained budget value since there are no rigid classification patterns. For instance, determining the detailed budget value seems challenging in the case of client 1 in SST-5. Therefore, developing an efficient method to infer the exact budgets based on these broad patterns for each client are required.

## 3.3 Observation Summary

In summary, our findings and the approach for designing an algorithm are as follows: (1) non-IIDness significantly affects the distributed ICL setting, necessitating the development of a coalition method. To handle this problem, it is straightforward to allocate an appropriate number of budgets to each client, *i.e.,* making server work so as it knows client all samples. (2) By analyzing the query embeddings, we can determine the importance of each client per query.

## 4 Method

In this section, we outline the proposed algorithm to mitigate non-IIDness in the ICL framework. Specifically, we show how to train the ***budget allocator*** and conduct inference.

---

**Algorithm 1** Construct dataset

**Require:** Encoder $\mathcal{E}(\cdot)$, server-side ICE budget $k$, proxy dataset $\mathcal{D}_{\text{proxy}} = \{(x_j, y_j)\}_{j=1}^{N_{\text{proxy}}}$ Quantization parameter $\delta$.
1: **for** $(x_j^{\text{proxy}}, y_j^{\text{proxy}}) \in \mathcal{D}_{\text{proxy}}$ **do**
2:    $e_j^{\text{proxy}} = \mathcal{E}(x_j^{\text{proxy}})$
        /* Get distributed examples */
3:    **for** $c \in [C]$ **do**
4:        $e_j^{\text{proxy}} \to$ Client $c$
5:        $S_c = \mathcal{T}(e_j^{\text{proxy}}, k | \mathcal{D}_c)$
6:        Server $\leftarrow S_c$
7:    **end for**
        /* Construct optimal example */
8:    $S = \bigcup_{c=1}^{C} S_c$
9:    $S^{\text{top}} = \underset{(x_s, y_s) \in \mathcal{S}}{\arg \text{Top-}k} \|e_j^{\text{proxy}} - \mathcal{E}(x_s)\|_2$
        /* Compute proper budget size for each $c$ */
10:    $k_c(e_j) = |S^{\text{top}} \cap S_c| // \delta \quad \forall c \in [C]$
11: **end for**
12: $B_{\text{proxy}} = \{(e_j, \{k_c(e_j)\}_{c=1}^{C})\}_{j=1}^{N_{\text{proxy}}}$
13: **return** $B_{\text{proxy}}$

---

**Algorithm 2** Top-$k$ sampling, $\mathcal{T}(e, k | \mathcal{D})$

**Require:** Query embedding $e$, Encoder $\mathcal{E}(\cdot)$
        /* Compute embedding */
1: **for** $(x_i, y_i) \in \mathcal{D}$ **do;** $e_i \leftarrow \mathcal{E}(x)$ **end for**
        /* Select top-$k$ samples */
2: $S = \underset{(x_i, y_i) \in \mathcal{D}}{\arg \text{Top-}k} \|e - e_i\|_2$
3: **return** $S$

---

**Algorithm 3** Inference (Client, Server)

**Require:** Embedding model $\mathcal{E}(\cdot)$, LLM $\mathcal{M}(\cdot)$, local datasets $\mathcal{D}_c$, budget allocator $f_c(\cdot)$ Buffering parameter $\alpha$.
**Input:** Test query $x_q$
1: Extract embedding $e_q = \mathcal{E}(x_q)$
2: **for** $c \in [C]$ **do**
3:    $\hat{k}_c = f_c(e_q)$
4:    Send $e_q$ to all clients
5:    $S_c = \mathcal{T}(e_q, \hat{k}_c + \alpha | \mathcal{D}_c)$
6:    return back $S_c \to$ Server
7: **end for**
8: $S = \mathcal{T}(e_q, k | \bigcup_{c \in [C]} S_c)$
9: $s(S, x_q) = (x_1, y_1) \odot ... \odot (x_k, y_k) \odot x_q$
10: **return** $y = \mathcal{M}(s(S, x_q))$

---

## 4.1 Train a Budget Allocator

Based on Section 3, it is feasible to assign budgets of each client by using the embeddings obtained from the retriever encoder $\mathcal{E}$. We first construct the datasets having the targeting budget values and then train the budget allocator. The pseudo-codes are described in Algorithm 1 and 2.

**Construct dataset for oracle budget.** First, we explain how to create a dataset to train the budget allocator for each client, as described in Algorithm 1. Given proxy dataset $\mathcal{D}_{\text{proxy}}$, for all embeddings $e_j = \mathcal{E}(x_j)$ where $(x_j, y_j) \in \mathcal{D}_{\text{proxy}}$, we request $k$ samples from each client $c \in [C]$ using Top-$k$ procedure, *i.e.,* $S_c = \mathcal{T}(e, k | \mathcal{D}_c)$. Once the server receives $k$ examples from each clients, *i.e.,* $\{S_c\}_{c=1}^{C}$, it merges and re-orders them to obtains $S^{\text{top}}$. Based on $S^{\text{top}}$, we count the number of samples from each client in $S^{\text{top}}$, *i.e.,* compute $k_c(e_j)$. After counting $k_c(e_j)$ for all clients, we quantize the budget

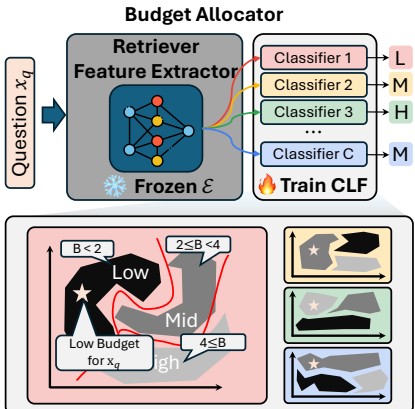

Figure 5: Overview of the budget allocator: We train a budget allocator on top of the frozen feature extractor $\mathcal{E}$, which inherits from the retriever. During inference, when a test query $x_q$ is provided, this module determines the quantized budget levels for each client and allocates them accordingly.

levels for each client using the quantization hyper-parameter $\delta$. As a result, the output of this procedure is $B_{\text{proxy}}$ for all clients, composed of embeddings $e$ and their respective budgets $k_c(e_j)$.

**Train budget allocator.** Based on the constructed dataset $B_{\text{proxy}}$, we train the *budget allcoators*, *i.e.,* $\{f_c(\cdot)\}_{c=1}^C$, for each $f_c(\cdot)$ has Multi-layer perceptrons on top of the frozen feature extractor of the off-the-shelf retriever $\mathcal{E}$. The budget allcoators are trained on the cross-entropy loss, as we have already quantized the optimal budgets using the hyper-parameter $\delta$. Note that if $\delta$ is high, the quantization is severe, otherwise the quantization is mild.

### 4.2 INFERENCE USING BUDGET ALLOCATOR

We derive the response to the test query $x_q$ utilizing the LLM $\mathcal{M}(\cdot)$ through the described steps (see Algorithm 3 for specifics). We first extract the embedding $e_q = \mathcal{E}(x_q)$. Then, we compute the allocated budget $\{\hat{k}_c = f_c(e_q)\}_{c=1}^C$ and send $\hat{k}_c$ to each client. Each client sends back top $\hat{k}_c + \alpha$ samples, *i.e.,* $S_c$, to the server. Note that we summarize how the budget allocator outputs $\hat{k}_c$ in Figure 5. Here, $\alpha$ denotes the buffering hyper-parameter, which increases the chances for each client to be involved. After collecting $S_{\text{agg}} = \bigcup_{c \in [C]} S_c$, we aggregate them and run the usual ICL procedure.

## 5 EXPERIMENT

### 5.1 EXPERIMENT SETUP

First, we summarize the baselines, datasets, and the method for constructing non-IID settings. Finally, we depict the implementation details.

**Baselines.** We compare our algorithm with various baselines, including social learning (Mohtashami et al., 2023), which does not account for non-IIDness, and other possible ways for handling distributed non-IID ICL, such as Zero-shot, Proxy-only, Singleton (single client), Uniform-budget, Random-budget, and $\infty$-budget (oracle case). The detailed explanations are described in Appendix C.

**Datasets.** We check the performance under 7 datasets – Sentiment classification: SST-5 (Socher et al., 2013), Amazon (McAuley & Leskovec, 2013), Yelp (Zhang et al., 2015), MR (Pang & Lee, 2005), Topic classification: Yahoo, AGNews (Zhang et al., 2015), and Subjectivity classification: Subj (Pang & Lee, 2004).

**Dataset partition for non-IIDness.** We split the training dataset into $C$ subsets to ensure they follow a non-IID distribution. To achieve this, we partition the data based on class, following the splitting criteria outlined in Li et al. (2022). Specifically, each client has access to only $\gamma < \Gamma$ classes, where $\Gamma$ represents the total number of classes. We outline the summary of $\gamma$ for each dataset in Appendix D.

| Algorithm | Dataset | | | | | | | Avg |
|---|---|---|---|---|---|---|---|---|
| | SST-5 | Amazon | Yelp | MR | Yahoo | AGNews | Subj | |
| Zero-shot | 29.19 | 24.70 | 31.23 | 73.95 | 25.87 | 67.60 | 50.55 | 43.30 |
| Proxy-only | 40.64± 2.89 | 28.43± 0.11 | 31.85± 1.28 | 70.40± 1.54 | 54.73± 0.93 | 84.65± 0.42 | 71.09± 1.34 | 54.54 |
| Singleton | 25.14± 4.18 | 24.03± 0.57 | 29.44± 3.91 | 50.00± 0.00 | 38.14± 2.03 | 50.60± 0.66 | 50.00± 0.00 | 38.19 |
| Social Learning | 36.03± 0.27 | 28.42± 0.19 | 29.25± 0.45 | 58.58± 0.18 | 46.03± 0.49 | 81.10± 0.29 | 71.37± 0.71 | 50.11 |
| Uniform-budget | 32.94 | 25.63 | 26.60 | 33.65 | 43.00 | 73.17 | 63.20 | 42.60 |
| Random-budget | 32.82± 0.82 | 25.69± 0.55 | 27.72± 0.51 | 34.68± 0.59 | 42.46± 0.53 | 67.34± 0.39 | 65.37± 0.80 | 42.30 |
| $\infty$-budget | 43.26 | 32.70 | 34.80 | 77.20 | 62.67 | 89.37 | 91.4 | 61.62 |
| **Ours** | **44.08± 0.12** | **31.54± 0.22** | **35.48± 0.28** | **80.44± 0.67** | **61.67± 0.25** | **88.52± 0.30** | **82.36± 0.91** | **60.58** |

Table 1: Main results: To address the issue of non-IIDness in distributed ICL, we examined seven datasets and seven straightforward baselines. We run three random seeds and illustrate mean and std values. The top performance is highlighted in **bold** font, excluding the infinite budget scenario due to its impracticality. In summary, the proposed method effectively mitigates the non-iid distributed ICL problem to a reasonable extent.

**Dataset paraphrasing.** Due to concerns about sharing private samples between servers and clients, various techniques have been developed for natural language tasks. In this paper, we adopt the paraphrasing technique used in Mohtashami et al. (2023). Specifically, we utilize a small language model (Team et al., 2024), designed for small terminal devices, to generate paraphrased questions. In Appendix E, we summarize the instructions provided to the language model for rephrasing queries in the training dataset.

**Implementation details.** We implement our method as well as baselines based on OpenICL (Wu et al., 2023). For the retriever scenario, we utilize the pre-trained KATE retriever (Liu et al., 2021), which has been trained on the SNLI (Young et al., 2014) and MultiNLI (Williams et al., 2018) datasets. Note that they do not overlap with the datasets used in our experiment. They used RoBERTa-large (Liu et al., 2019) encoder model. We use GPT-Neo-2.7B (Black et al., 2021) pre-trained model as answering LLMs as default. Hyper-parameters related to training budget allocators, $\alpha$, and $\delta$ are described in Appendix D in detail.

## 5.2 MAIN RESULTS

We have presented the performance of our algorithm and baselines in Table 1. First, we can observe that performance varies significantly depending on the way the budget is allocated, which indicates that the budget allocation scheme really matters in distributed non-IID ICL. Additionally, even when using only the proxy dataset, there is a performance improvement, and this performance surpasses that of using other clients which have the tilted local datasets (*e.g.,* $29.19\% \rightarrow 40.64\%$ in SST-5 case). This indicates that utilizing a biased dataset can degrade the ICL performance. Although social learning algorithm has shown good performance in the previous paper, it does not perform well under the non-IID cases configured in this research. If we can use an infinite budget, all settings would exhibit high performance. However, our proposed algorithm demonstrates better performance than the infinite budget upper limit (*e.g.,* $34.86\% \rightarrow 35.48\%$ in the Yelp case). This is likely due to a mechanism that prevents unnecessary information from being selected by the retriever with high importance. Ultimately, the proposed algorithm shows an average performance improvement of $5.05\%$ across seven datasets compared to the best performance of baselines using the proxy dataset. This shows that the proposed algorithm can handle the non-IID case well.

## 5.3 ANALYSIS

In this section, we further examine four key aspects: (1) privacy-preserving case analysis, which encompasses paraphrasing both training and testing queries, (2) sensitivity to hyper-parameters, (3) the performance of the trained budget allocator, and (4) the compatibility of the LLMs.

**Paraphrasing results.** Due to privacy concerns in the fundamental distributed system, we evaluate the performance of paraphrased datasets, with results detailed in Table 2. Our method demonstrates superior performance compared to other baselines across multiple datasets. We used the exact same data settings as in Table 1. Specifically, performance on the Subj and SST-5 datasets is lower than

| Algorithm | Dataset | | | Avg |
|---|---|---|---|---|
| | SST-5 | Yelp | Subj | |
| Zero-shot | 27.96 | 31.40 | 51.55 | 36.97 |
| Proxy-only | 39.39± 1.33 | 31.78± 1.75 | 73.46± 1.46 | 48.21 |
| Singleton | 25.31± 3.89 | 30.78± 4.88 | 50.08± 0.10 | 35.39 |
| Social Learning | 33.09± 0.68 | 28.80± 0.33 | 74.82± 0.93 | 45.47 |
| Uniform-budget | 27.06 | 26.60 | 63.30 | 38.99 |
| Random-budget | 27.29± 0.51 | 27.70± 0.46 | 63.88± 0.81 | 39.62 |
| $\infty$-budget | 41.63 | 37.23 | 90.75 | 56.54 |
| Ours | **40.37**± 0.27 | **36.52**± 0.89 | **83.82**± 1.00 | **53.57** |

Table 2: Analysis of the generated query and training samples. We paraphrase the datasets using small-sized LLMs and conduct the experiments as in Table 1 under the same experimental settings.

without paraphrasing, while the Yelp dataset shows a slight improvement. Additionally, as consistent with Table 1, non-IIDness causes significant performance degradation for ICL methods, as seen by comparing Zero-shot with ICL-related methods (*e.g.,* $27.96\% \rightarrow 25.31\%$ in the Singleton case).

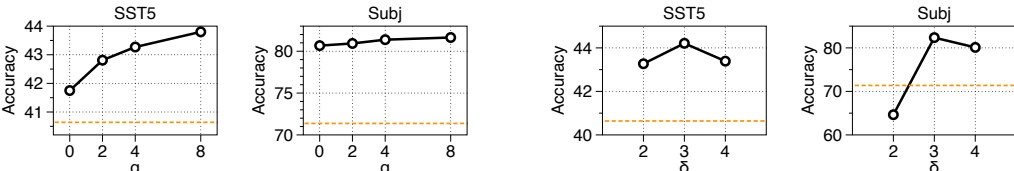

Figure 6: Additional budget $\alpha$ analysis. The orange dash line is the second-best baseline.

Figure 7: Budget allocator resolution $\delta$ analysis. The orange dash line is the second-best baseline.

**Hyper-parameter sensitivity.** We examine the sensitivity of the hyper-parameters of our method. We have two hyper-parameters: $\delta$, which is the resolution of the budget allocator; $\alpha$, which represents the additional budget allocated to each client as a buffer; and proxy size, which is the size of proxy data for the budget allocator training. As illustrated in Figure 7, when we increase $\alpha$, the performance is improved while the budget efficiency is reduced. On the other hand, when $\delta$ is high (or low), it has too dense (or sparse) representation of the budget class, thus performance is degraded. Nevertheless, the performance is higher than the other baselines in Table 1. For the sensitivity of the size of proxy data, it is revealed that our framework is not sensitive to how many proxy data samples are used to train the budget allocator, as shown in Figure 8. This indicates our method is stable even with limited proxy data on the server side.

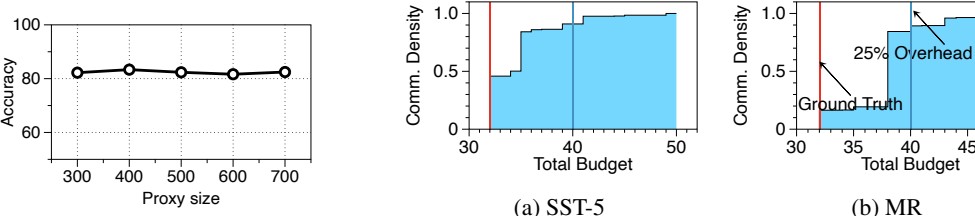

Figure 8: Proxy size analysis. We check Subj dataset under `GPT-Neo-2.7B`.

Figure 9: Analyze the allocated budget. Analyze the total amount of budget allocated to clients under two datasets. Red and blue lines denote the oracle and $25\%$ larger total budgets compared to the oracle case.

**Trained budget allocator.** We assess whether the trained budget allocator distributes budgets appropriately for each client. To evaluate efficiency, we examine the number of samples, *i.e.,* $\hat{k}_c$ communicated for all queries and plot a histogram. As demonstrated in Figure 9, we confirm that the proposed algorithm's forecasts exhibit nearly identical performance to the oracle budget when

an additional 25% budget is allocated. Note that without the proposed algorithm, it is necessary to assign $k \times C$ number of budgets to get a performance similar to the oracle case.

| Algorithm | Architecture | | | |
|---|---|---|---|---|
| | GPT-Neo-1.3B | GPT-Neo-2.7B | Llama-2-7B | gpt-3.5-turbo |
| Zero-shot | 51.30 | 50.55 | 49.10 | 57.57 |
| Proxy-only | 80.18± 1.87 | 71.09± 1.34 | 88.13± 0.74 | 88.44± 0.69 |
| Singleton | 50.00± 0.00 | 50.00± 0.00 | 52.89± 3.43 | 60.81± 6.31 |
| Social Learning | 68.55± 0.64 | 71.37± 0.71 | 88.82± 0.50 | 87.53± 0.46 |
| Uniform-budget | 44.40 | 63.20 | 54.00 | 81.23 |
| Random-budget | 43.68± 0.80 | 65.37± 0.80 | 55.60± 0.41 | 81.47± 1.81 |
| $\infty$-budget | 92.05 | 91.40 | 92.30 | 92.23 |
| Ours | **85.73**± 0.94 | **82.36± 0.91** | **91.58**± 0.14 | **91.33**± 0.72 |

Table 3: Default non-IID setting of Subj using different LLMs. 32 ICEs for server LLM inference.

**Other types of LLMs.** We utilize various LLM architectures to assess the compatibility of the proposed algorithm. Specifically, we evaluate the SST-5 dataset using different model sizes, including GPT-Neo-1.3B (Black et al., 2021), Llama-2-7B (Touvron et al., 2023a), and the OpenAI gpt-3.5-turbo (OpenAI, 2022). As demonstrated in Table 3, our method exhibits a plug-and-play capability and achieves reasonable performance improvements in the distributed non-IID ICL.

## 6 RELATED WORK

**In-context learning.** ICL (Dong et al., 2022) is one of the fastest paradigms using pre-trained LLMs by feeding several examples to construct the context to solve the given query. The main criteria of this research field are to find the most informative samples among the training datasets. For example, Liu et al. (2021) trains BERT (Devlin et al., 2018) oriented encoder and uses the $k$ nearest neighbors. One of the reasonable sparse retriever, rule-based approaches is using BM25 (Robertson et al., 2009), which measures the term-frequency. Rubin et al. (2022) proposed an efficient retriever called EPR. It trains two encoders by inheriting the method of dense passage retriever (DPR) (Karpukhin et al., 2020) under the loss of positive and negative pairs. To reduce the domain specificity, Li et al. (2023) proposed UDR, which is applicable to multiple domain tasks in a universal way and shows reasonable performance from a single retriever. PromptPG (Lu et al., 2022) utilized a reinforcement learning framework to train the retriever so that it can generate context to improve the answerability of LLMs. Similarly, LLM-R (Wang et al., 2023) uses a reward model to train the retriever. Chang & Jia (2022) trains linear regressors according to the example influence on the LLM prediction. Xie et al. (2021) proposes to use implicit Bayesian inference to understand the ICL problem. Mavromatis et al. (2023) proposes AdaICL to handle the efficient ICL with a limited annotation budget. Note that extensive research focuses on the centralized case rather than targeting distributed cases.

**Distributed ICL.** To the best of our knowledge, only a single study (Mohtashami et al., 2023) tries to address ICL in a distributed manner. However, this paper solely focuses on merging the distributed information without considering the nature of the non-identically distributed information. Many studies, such as those on federated learning (Li et al., 2021; Zhang et al., 2021; Mammen, 2021), address the non-IID distribution of datasets, highlighting the need to handle distributed non-IID ICL.

## 7 CONCLUSION

In this paper, we tackle the challenge of ICL when datasets are distributed among clients with non-IID. Initially, we examine if non-IID leads to performance degradation and discover that they cause significant drops in performance. Inspired by the learnable pattern between budget values and query embeddings, we propose an algorithm that learns the task of budget assignment and employs it during inference to allocate appropriate budgets for each query. Using this proposed algorithm, we achieve performance improvements across several benchmarks compared with various baselines. In addition, we examine the privacy-preserving version of our method using paraphrasing technology and show its effecacy. Last but not least, extensive sensitivity experiments show the robustness of our method on hyper-parameters and different LLMs.

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

---

-Supplementary Material-

# Distributed In-Context Learning under Non-IID Among Clients

---

This is the supplementary material for the Distributed In-Context Learning under Non-IID Among Clientspaper. Due to page limitations, we provide additional details as follows: (1) A detailed description of constructing non-IIDness in Appendix A, Following that, we outline the experimental setting of Section 3 in Appendix B. In Appendix C and Appendix D, we describe the baselines of Table 1 and Section 5, respectively. Lastly, we summarize the method of constructing generated samples for privacy in Appendix E.

## A   HOW WE CONSTRUCT NON-IIDNESS

Following Li et al. (2022), we use class number based non-IID partition in our experiment. For a dataset with overall $\Gamma$ classes, given hyperparameter class number $\gamma$ on each client, we randomly assign $\gamma$ classes from the overall $\Gamma$ classes for each client. Assuming that $C_1 \leq C$ clients are assigned with a specific class, we equally partition samples of this class into $C_1$ parts and assign one part to each of $C_1$ clients. We denote this non-IID partition with the class number $\gamma$ on each client as `noniid-#label=`$\gamma$.

## B   MOTIVATION EXPERIMENTAL SETTINGS

**Non-IIDness performance drop experiment.**  For this experiment, we use SST5, Amazon, Yelp, Yahoo, and AGNews. And the non-iid settings are dsecribed in Table 4.

| Dataset | ICE size $k$ | #Clients | Partition |
|---------|-----------|----------|-----------|
| SST-5 | 32 | 8 | `noniid-#label=1` |
| Amazon | 16 | 8 | `noniid-#label=1` |
| Yahoo | 16 | 8 | `noniid-#label=2` |
| AGNews | 16 | 4 | `noniid-#label=1` |
| Yelp | 4 | 2 | `noniid-#label=3` |

Table 4: Experimental setup for obtaining the motivation.

**t-SNE analysis of per-client budget experiment.**  For extracting t-SNE figure, we utilized the following experimental setting Table 5

| Dataset | ICE size $k$ | #Clients | Partition |
|---------|-----------|----------|-----------|
| SST-5 | 32 | 4 | `noniid-#label=2` |
| Yelp | 8 | 2 | `noniid-#label=3` |

Table 5: Experimental setup for obtaining the motivation.

## C    BASELINE DETAILS

**Proxy-only.** We randomly select samples from the original test set to construct the proxy set on the server side and use the remaining test set as the true test set. When performing the ICL process, the server directly retrieves ICEs from the proxy set rather than from the training set. For SST5, MR, and Subj, we randomly select 500 samples from the test set to be the proxy set. For Amazon, Yelp, Yahoo, and Agnews, we randomly select 750 samples from the test set to be the proxy set. Also, since the proxy set is already on the server side, there will be no privacy issues during communication between clients and the server. Thus, we don't generate samples to protect privacy and directly use the original samples in the proxy set for ICL.

**Singleton.** This baseline is for if the whole ICE set is constructed only using single client's local dataset. We randomly select one client from $C$ clients, and perform local retrieval with $k_c = k$ budget. Then, the server uses this locally retrieved ICE set for LLM inference. We report the average accuracy over all clients.

**Social learning.** This algorithm Mohtashami et al. (2023) is the first paper that considers the distributed ICL, but it only considers the IID setting. Since the authors didn't release the source code, we implemented it on our own. In our implementation, given server-side ICE number as $k$, each local client $c$ performs local top-$\lceil \frac{k}{C} \rceil$ retrieval and sends retrieved ICEs to the server. The server then performs a random selection from $k$ ICEs to construct an ICE set with $k$ samples and feed this ICE set into LLM for inference.

**Uniform-budget.** We equally assign a local budget to each client. Assume the ICE number fed to server-side LLM for inference is $k$, then each client's local budget is $\lceil \frac{k}{C} \rceil$, where $C$ is the number of clients. On server-side aggregation, we use `reorder` method as default.

**Random-budget.** We randomly assign a local budget to each client with the constraint that the overall local budget over $C$ clients is $k$, where $k$ is the ICE number fed to server-side LLM. On server-side aggregation, we use `reorder` method as default.

**∞-budget.** The most inefficient way to do distributed non-IID ICL is to allow ∞-budget on each client, that is, sending all samples to the server side. Then, the system performs centralized retrieval on the collected dataset to obtain top-$k$ ICEs and feed them into LLM for inference.

## D    EXPERIMENTAL SETTING

**Dataset Explanation.** In this study, we utilized seven text classification tasks: four for sentiment analysis, two for topic classification, and one for subjectivity classification. The dataset statistics are presented in Table 6.

| Dataset | Type | Training | Test | Class |
|---------|------|----------|------|-------|
| SST-5 | Sentiment | 8.534 | 2,210 | 5 |
| Amazon | Sentiment | 30,000 | 3,000 | 5 |
| Yelp | Sentiment | 30,000 | 3,000 | 5 |
| MR | Sentiment | 8,662 | 2,000 | 2 |
| Yahoo | Topic | 29,150 | 3,000 | 10 |
| AGNews | Topic | 29,914 | 3,000 | 4 |
| Subj | Subjectivity | 8,000 | 2,000 | 2 |

Table 6: The statistics of the datasets used.

Given that the input instruction prompt can notably influence performance, we detail the prompts used for each dataset in Table 17. It is in the last page since prompts have long length. We follow the prompt settings described in Li et al. (2023) and use the dataset uploaded by the paper's author, available at https://huggingface.co/KaiLv.

**ICE number for LLM inference.** Given an LLM, different datasets show different preferences on the choice of ICE number, *i.e.,* $k$, used in ICL inference for better performance. For algorithms using ICL (except Zero-shot), SST5, MR, and Subj use 32 ICEs for server-side LLM inference; Amazon

uses 8 ICEs for server-side LLM inference; Yelp, Yahoo, and Agnews use 4 ICEs for server-side LLM inference.

**Non-IID Setting.** To keep similar non-IIDness levels across different datasets, we follow Table 7 as non-IID hyper-parameters for each dataset.

**Hyper-parameters for our methods.** For the main table results, the generated dataset results, and the different LLM architecture results, the hyper-parameters are shown in Table 8, Table 9 and Table 10, respectively. For the training of the budget model, we use 800 epochs, with a learning rate range $\{0.01, 0.003\}$ and a batch size of 8.

**Multi-layer perceptron for budget allocator.** We use the three-layer perceptron on top of the encoder $\mathcal{E}$. The torch pseudo code is as follows:

```
class SMLP(nn.Module):
    def __init__(self, width=300, num_classes=10,
data_shape=(768,)):
        super().__init__()
        self.flat = nn.Flatten()
        self.l1 = nn.Linear(np.prod(data_shape), width)
        self.relu = nn.ReLU()
        self.l2 = nn.Linear(width, width)
        self.l3 = nn.Linear(width, num_classes)

    def forward(self, x):
        x = self.flat(x)
        x = self.l1(x)
        x = self.relu(x)
        x = self.l2(x)
        x = self.relu(x)
        x = self.l3(x)
        x = F.softmax(x)
        return x
```

| Dataset | #Clients | Partition |
|---------|----------|-----------|
| SST-5 | 4 | noniid-#label=2 |
| Amazon | 2 | noniid-#label=3 |
| Yelp | 2 | noniid-#label=3 |
| MR | 4 | noniid-#label=1 |
| Yahoo | 2 | noniid-#label=5 |
| AGNews | 2 | noniid-#label=2 |
| Subj | 4 | noniid-#label=1 |

Table 7: Non-IID setting

| Dataset | ProxySetSize | $\delta$ | $\alpha$ | QuantRatio |
|---------|--------------|----------|----------|------------|
| SST-5 | 500 | 3 | 0 | 0.5 |
| Amazon | 750 | 2 | 0 | 0.5 |
| Yelp | 750 | 2 | 2 | 0.5 |
| MR | 500 | 3 | 0 | 0.5 |
| Yahoo | 750 | 2 | 2 | 0.5 |
| AGNews | 750 | 2 | 2 | 0.5 |
| Subj | 500 | 3 | 0 | 0.3 |

Table 8: Hyper-parameters of our methods used in the main table

| Dataset | ProxySetSize | $\delta$ | $\alpha$ | QuantRatio |
|---------|-------------|----------|----------|------------|
| SST-5 | 500 | 3 | 4 | 0.5 |
| Yelp | 750 | 3 | 0 | 0.5 |
| Subj | 500 | 3 | 0 | 0.3 |

Table 9: Hyper-parameters of our methods used for the generated query and training samples experiment

| Model | ProxySetSize | $\delta$ | $\alpha$ | QuantRatio |
|-------|-------------|----------|----------|------------|
| GPT-Neo-1.3B | 500 | 3 | 0 | 0.3 |
| GPT-Neo-2.7B | 500 | 3 | 0 | 0.3 |
| Llama-2-7B | 500 | 3 | 0 | 0.3 |
| gpt-3.5-turbo | 500 | 3 | 0 | 0.3 |

Table 10: Hyper-parameters of our methods used for different LLM architectures experiment on Subj

## E    GENERATE PARAPHRASED QUESTION

To generate the paraphrased query and response, we use the following instruction.

Please paraphrase the original sentence. Original sentence: {In-context example} Paraphrase sentence: {Paraphrased sentence}

Here is an example of input for the rephrasing LLM using the SST-5 dataset.

Please paraphrase the original sentence. Original sentence: "a stirring, funny and finally transporting re-imagining of beauty and the beast and 1930s horror films" Paraphrased sentence: A captivating, humorous, and ultimately uplifting reinterpretation of Beauty and the Beast combined with 1930s horror films. Please paraphrase the original sentence. Original sentence: "jonathan parker 's bartleby should have been the be-all-end-all of the modern-office anomie films" Paraphrased sentence: Jonathan Parker's "Bartleby" had the potential to be the definitive film capturing the sense of alienation in modern office settings. Please paraphrase the original sentence. Original sentence: "a fan film that for the uninitiated plays better on video with the sound turned down" Paraphrased sentence: A fan film that, for those not familiar with the source material, is more enjoyable when watched with the sound turned off. Please paraphrase the original sentence. Original sentence: "apparently reassembled from the cutting-room floor of any given daytime soap" Paraphrased sentence: It appears to be pieced together from the outtakes of any given daytime soap opera. Please paraphrase the original sentence. Original sentence: "" Paraphrased sentence:

Our paraphrased examples are summarized as follows.

| Original | Paraphrased |
|----------|-------------|
| a turgid little history lesson , humourless and dull | A dry and tedious history lesson that is devoid of humour or interest. |
| not so much a movie as a picture book for the big screen . | The movie is more of a picture book than a full-fledged movie for the big screen. |
| now it 's just tired . | It is now simply outdated. |

Table 11: Paraphrased examples of SST-5 dataset

| Original | Paraphrased |
|---|---|
| for all the wit and hoopla , festival in cannes offers rare insight into the structure of relationships . | Festival in Cannes offers rare insight into the structure of relationships. |
| eldom has a movie so closely matched the spirit of a man and his work . | A movie seldom has a movie so closely matched the spirit of a man and his work. |
| those of you who are not an eighth grade girl will most likely doze off during this one . | 8th graders and younger will most likely doze off during this one. |

Table 12: Paraphrased examples of Subj dataset

## F EXTRA EXPERIMENT

### F.1 ROBUSTNESS ON PROXY SIZE

Here, we present more detailed results on Subj with different proxy sizes over different values on budget allocator resolution $\delta$ in Figure 10.

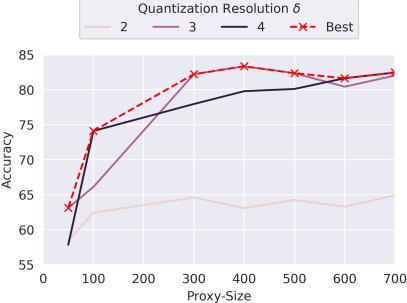

Figure 10: Proxy size robustness over different budget allocator resolution $\delta$. We check Subj dataset under `GPT-Neo-2.7B`.

### F.2 NON-EXTREME NON-IID ON BINARY CLASSIFICATION TASKS

We conduct the experiment on Dirichlet distribution $\mathrm{Dir}(\alpha)$ Non-IID partition on Subj and MR under the setting of 4 clients with $\alpha = 1.5$. The per-client sample distribution is shown in Figure 11, and the performance results are shown in Table 14.

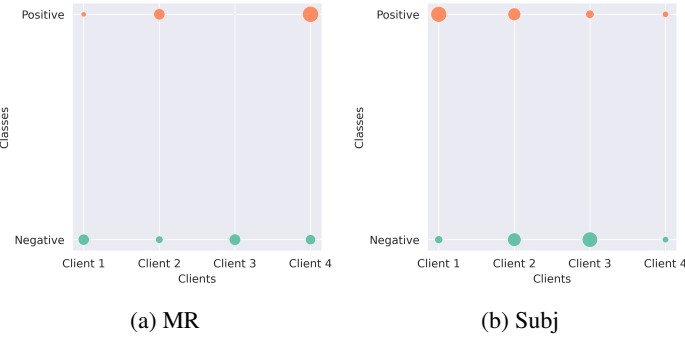

(a) MR  (b) Subj

Figure 11: Per-client sample distribution under Dirichlet distribution Non-IID setting.

| Original | Paraphrased |
|---|---|
| A friend of mine suggested we go here today before our movie. I was planning on suggesting another place, but she got their early and got a table DARN!I don't hate Red Robin...I think I avoid it because I am not a big fan of hamburgers. Seems like more of a place for straight guys and kids if you ask me, but my experience today wasn't to bad.Our waiter was really nice however I think that may have been a result of my push up bra.Ordered the Crispy Chicken Salad which also had hard boiled egg, bacon, and veggies and was very good. I'll probably get that again next time someone drags me here. Get it with that Italian dressing. Yummy! My friend ordered us onion rings as an appetizer I hate onions, but those were goood! Not my first choice but good crowd pleaser with more decent food than most chains. | I was considering going to Red Robin with a friend, but I decided to go somewhere else instead. I was not a big fan of hamburgers, but I was drawn to Red Robin because of the free tastings. I was pleasantly surprised by the friendliness of the staff, especially the waiter who was very attentive and helpful. I ordered the Crispy Chicken Salad, which was very good. I'll probably go back there next time someone drags me there. |
| Huge rooms. Soft towels. Comfortable bed with tons of pillows. Three things that make me happy when I'm traveling. What's wrong with this place, then? First of all, the location is central to nothing aside from the mall which houses several chain restaurants. Second, the staff could not care less about pretty much anything. While the woman who checked us in was helpful and friendly, that was the end of that. Despite requesting my Hilton status "amenities" (just a package of snacks and water...not much), I never received it. I was also charged $30 for internet when I shouldn't have paid a thing (got the bill corrected at check out). And housekeeping didn't leave enough towels or replace the water. Considering we paid over 200 a night, the blase attitude of the staff is very disappointing. | 3 things that make me happy when I travel are: spacious rooms, friendly staff, and a comfortable bed with lots of pillows. However, the location is inconvenient, the staff is indifferent, and the amenities are subpar. |
| You can either pay $5.50 for 3-day movie rentals or get a $40 membership and pay $2 for 3-day/$3 for 7-day rentals. They also offer 2-for-1 movies for students on Tuesday and Thursday. The reason I give 3 stars is because that deal isn't valid for people with memberships!? I learnt this 30 seconds after paying for a membership. I'm a student and could have paid $2.75 for movies twice a week. Instead I paid $40 (way too much) for a membership to pay .75 cents less than the other students. Good movie selection and shop but don't fall for their rip off of a membership unless you rent daily. | 3-day movie rentals cost either $5.50 or $40 for a membership. They offer a deal for students on Tuesdays and Thursdays, but it's not valid for those with memberships. |

Table 13: Paraphrased examples of Yelp dataset

## F.3    T-SNE UNDER NON-IID WITH TASK SHIFTING & FEATURE SKEW

Dataset Amazon and Yelp are 5-class sentiment classification tasks with exactly the same label space, while different text query distributions. Based on this, we design a special Non-IID setting with task shifting & feature skew between clients: client 1 only contains $10,000$ Amazon training samples,

| Algorithm | Dataset | |
|---|---|---|
| | MR | Subj |
| Zero-shot | 73.95 | 50.55 |
| Proxy-only | 70.40 | 71.09 |
| Singleton | 64.16 | 73.80 |
| Social Learning | 58.85 | 76.95 |
| Uniform-budget | 52.85 | 77.80 |
| Random-budget | 53.50 | 77.85 |
| $\infty$-budget | 77.20 | 91.40 |
| **Ours** | **75.53** | **82.80** |

Table 14: MR, Subj results under Dirichlet distribution Non-IID.

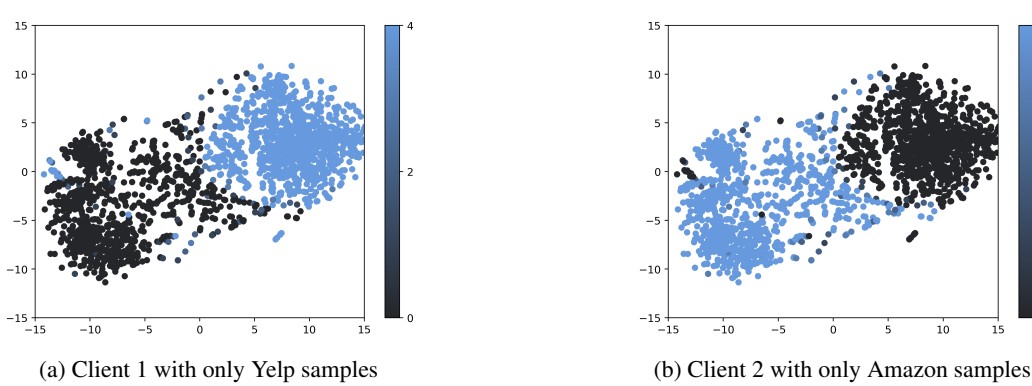

(a) Client 1 with only Yelp samples      (b) Client 2 with only Amazon samples

Figure 12: t-SNE analysis on the test set consisting of both Yelp & Amazon samples. Data points are colored based on local oracle budget values.

and client 2 only contains $10,000$ Yelp training samples. Thus, we consider this special setting to be a task-shifting Non-IID. Also, since each client consists of samples from all classes of each task, we consider this setting as feature-skew Non-IID with class balance. We calculate the oracle budget values for a mixed test set consisting of $1,000$ Yelp test samples and $1,000$ Amazon test samples. Then we perform t-SNE analysis on sample embeddings of this mixed test set, colored using oracle local budget values. As shown in Figure 12, under Non-IID with task shifting & feature skew, there still exists clear clustering pattern between query embedding and oracle budget values. This indicates our method still can work with task shifting and feature skew.

### F.4 DISTRIBUTION SHIFT BETWEEN PROXY SET AND TEST SET

It is critical to control the distribution shifting between proxy set and test set. We conduct experiments on two settings for proxy set distribution different from test set.

**From the same dataset but different label distribution.** The most simple case of "different distribution" can come from the different label distribution skew between proxy set and test set. For this setting, we experiment on Subj with a proxy set only containing samples of one class. As shown in the last line of Table 15, when the label skew exists, the performance of our method does decrease compared with the setting using the ideal proxy set (from $82.36\%$ to $70.17\%$). However, it is still higher than some baselines, like zero-shot, singleton, uniform-budget, random-budget.

**Similar task but different dataset.** A more extreme case for proxy set different from test set can be, proxy set share same task with the test set, but are from different datasets. For this setting, we conduct the following experiment:

- use Amazon as proxy set for Yelp Non-IID setting (evaluate on Yelp test set)

- use Yelp as proxy set for Amazon Non-IID setting (evaluate on Amazon test set)

Since Yelp and Amazon share the similar task, we can consider this setting as using available dataset with similar task with the test set to construct the proxy set. We present the result in the the last line in Table 16. It shows that for the Amazon setting, using Yelp as a proxy set, the performance drop of our method is slight, and our method still outperforms other baselines, except in the ideal case where we use Amazon samples as a proxy set. While for Yelp setting using Amazon as proxy set, our method surprisingly shows even better performance than the ideal case, where use Yelp as proxy set.

| Algorithm | Subj |
|---|---|
| Zero-shot | 50.55 |
| Proxy-only | 71.09 |
| Singleton | 50.00 |
| Social Learning | 71.37 |
| Uniform-budget | 63.20 |
| Random-budget | 65.37 |
| ∞-budget | 91.30 |
| **Ours** | **82.36** |
| *Ours-proxy-label-skew* | *70.17* |

Table 15: Comparison with proxy set with label skew compared to the test set. The last line is the performance for this setting.

| Algorithm | Dataset | |
|---|---|---|
| | Amazon | Yelp |
| Zero-shot | 24.70 | 31.23 |
| Proxy-only | 28.43 | 31.85 |
| Singleton | 24.03 | 29.44 |
| Social Learning | 28.42 | 29.25 |
| Uniform-budget | 25.63 | 26.60 |
| Random-budget | 25.69 | 27.72 |
| ∞-budget | 32.70 | 34.80 |
| **Ours** | **31.54** | **35.48** |
| *Ours-diff-proxy* | *31.27* | *37.33* |

Table 16: Comparison with using different dataset to construct proxy set for budget allocator training. The last line is the performance for this setting.

## G    CONCRETE EXAMPLE OF DISTRIBUTED NON-IID ICL SCENARIO

A concrete example of distributed Non-IID ICL scenario can be the medical diagnosis task based on ICL cooperating with multiple medical institutions. Now, we have several medical institutions, with each institution owning some medical records (each sample consisting of the patient's symptoms description in text and the corresponding diagnosed disease, that is, the query $x$ and label $y$). These medical institutions normally do not have enough local computation power to support LLM computation requiring large GPU resources, while they can do some small-cost local computation like retrieval processes to find similar queries. At the same time, there will be a platform operating like a server in this system, with enough computation resources to support LLM inference and in charge of cooperation management between these institutions. Once the system (including the server platform and cooperating institutions) is deployed, the platform can provide consulting diagnosis services to other patients, doctors, or even other medical institutions based on pay-by-use knowledge pricing strategies. That is, the price is decided by the number of samples involved in the whole diagnosis procedure. Also, due to medical privacy concerns, the server platform can use local samples to perform inference while not allow caching these samples. Thus, these local retrieved samples cannot be cached to construct a retrieval pool on server platform. For the specific example of platform that supports LLM, OpenAI now provides ChatGPT Enterprise [3], which allows the deployment requirement that the platform should not cache and utilize private data for further training.

## H    DISCUSSION ON RELATION WITH DISTRIBUTED RAG

Here, we discuss the differences between our approach and existing distributed RAG studies to provide additional clarity and context for our contribution.

In developing this work, we carefully considered related studies in distributed RAG. However, the challenges addressed by existing distributed RAG works differ from those tackled in our paper. For instance, Wang et al. (2024) focuses on the creation of datasets for distributed RAG frameworks and explores LLM-based labeling techniques for engineering pipelines. Their research scope and methodology are distinct from ours and are not directly applicable to our specific problem setting. Similarly, Li et al. (2024) addresses resource consumption and real-time response challenges in distributed RAG, emphasizing local retrieval efficiency and answer accuracy. However, it does not account for the non-IID property in distributed settings. Additionally, Li et al. (2024) permits LLM deployment on partial local institutions, which is fundamentally different from our setting.

Real-world distributed non-IID RAG scenarios present a more complex framework involving numerous challenges that must be addressed for effective deployment. For example:

- How can we effectively decompose a user query into subqueries while considering local knowledge distribution?
- What is the best way to assign these subqueries to clients with varying local expertise?
- How should we merge knowledge retrieved from multiple clients with overlapping expertise, and should we assign confidence levels to different clients for the same subqueries?
- How can the local retrieval process be accelerated when dealing with large local databases?

These challenges represent broader avenues for exploration in distributed non-IID RAG. While our current work cannot be directly compared with existing distributed RAG studies due to different settings, we believe it offers an interesting starting point for addressing such challenges. Specifically, our approach focuses on how to enable cooperation among clients with varying distributions of knowledge. By assigning preferences to clients based on their local knowledge distributions and employing an MLP to learn these distributions without transmitting complete local knowledge to a central server, we offer an intuitive method that could inspire future advancements in distributed non-IID RAG.

---

[3] https://openai.com/enterprise-privacy/

| Dataset | Prompt | Label | Label Template | Example |
|---|---|---|---|---|
| AGNews | Topic of the text: | {World, Sports, Business, Technology } | Topic: *Label* | REDMOND, Wash. - Microsoft Corp. and cable television provider Comcast Corp. said Monday they would begin deploying set-top boxes powered by Microsoft software starting next week. \n Topic: Business \\...Oil demand is rising faster than predicted this year as OPEC pumps more low-quality oil in a failed bid to reduce record prices, according to International Energy Agency, an adviser to 26 industrialized nations. \n Topic: |
| MR | Sentiment of the sentence: | {great, terrible} | It was *Label* | "Analyze That" is one of those crass, contrived sequels that not only fails on its own but makes you second-guess your affection for the original. \n It was terrible ...about the only thing to give the movie points for is bravado—to take an entirely stale concept and push it through the audience's meat grinder one more time.\n It was |
| SST-5 | Sentiment of the sentence: | {great, good, okay, bad, terrible} | It was *Label* | a strong, funny, and finally transporting re-imagining of Beauty and the Beast and 1930s horror films \n It was great ...no movement, no yuks, not much of anything. \n It was |
| Subj | Subjectivity of the sentence: | {subjective, objective} | It's *Label* | gangs, despite the gravity of its subject matter, is often as fun to watch as a good spaghetti western. \n It's subjective ...smart and alert, Thirteen Conversations About One Thing is a small gem. \n It's |
| Amazon | Sentiment of the sentence: | {great, good, okay, bad, terrible} | It was *Label* | Love the originality of this music. Because she is ever-changing, Madonna is never boring. "Music" makes you want to dance - totally energizing! Wish the "Music" video was half as impressive as this work of art. ... The case for the DVDs were a bit damaged. The damage did not compromise the DVDs, ... \n It's |
| Yelp | Subjectivity of the sentence: | {subjective, objective} | It's *Label* | My family visited ceasars palace and ate here. Our waiting time was only ten minutes despite all the people. Our server was the best. He recommended several great dishes. The food was higher than our expections....his place is great. The staff is really friendly and the chile verde burrito is fantastic. You know a ... \n It's |
| Yahoo | Topic of the sentence | {Society & Culture, Science & Mathematics, Health, Education & Reference, Computers & Internet, Sports, Business & Finance, Entertainment & Music, Family & Relationships, Politics & Government} | It's *Label* | who sang message in a bottle? Answer: Sting (the Police) ...Neuroscience Question? Answer: no it is callef motor neuro-prostheses. \n Topic: |

Table 17: Prompt and instructions used for each dataset. We denote examples in blue and queries in red.

