# OpenReview forum: "Distributed In-Context Learning under Non-IID Among Clients"
_ICLR.cc/2025/Conference — Submitted to ICLR 2025_

### Official Review · Reviewer_GvJK · 2024-10-27

**Soundness:** 3
**Presentation:** 3
**Contribution:** 2
**Rating:** 6
**Confidence:** 4

**Summary:**

This paper proposed a solution to ICL when non-iid data exist across clients. The authors identify that traditional ICL assumes a centralized and uniform dataset that might not hold in real-world use cases (distributed), where each client's data might vary significantly. Thi could lead to suboptimal results by uniformly retrieving ICEs. In this paper the authors propose a framework that trains a budget allocator to determine the optimal number of ICEs to retrieve from each client based on their data's relevance to the query. The allocator uses a server-side proxy dataset to guide budget assignment per query - and this adjusts the contributions from each client and therefore more relevant context for inference. The experiments in this paper cover various LLM architectures over a series of classification tasks and demonstrate that this method improves ICL performance over existing distributed strategies (especially in non-iid scenarios)

**Strengths:**

This paper identifies a real-world complexity which is the non-iidess of the dta settings during ICL. It introduced a budge allocator that dynamically selects the most relevant ICEs from each client based on their own queries, which leads to improvement over uniform or random ICE distributions.  It also includes a privacy-preserving option using paraphrasing and demonstrates consistent performance gains over such privacy-preserving setting.

**Weaknesses:**

1. I feel the task covered in the study is quite limited to classification tasks. It would benefit from proving the effectivenss across other more complex tasks that are more context-heavy such as RAG and multihop reasoning.
2. The paraphrasing based method to secure privacy during data retrieval - seems to have limited evaluation and more robust evaluation against other privacy-preserving techniques needs to be done to support the claim
3. The proposed solution relies on the assumption that a high quality, representative proxy dataset presents at server-side. Although experiments have shown the method stability data size. it would be nice to see furtehr experiments on how the proxy data quality or distribution affects performance or alternative methods to reduce such dependency.

**Questions:**

1. In a real world use case, even when sharing a similar task, the clients might have totally different prompts in terms of structure, length, and specific requirements. Will this affect retrieval effectiveness since it heavily depends on the similarity between query and training examples?
2.  Could the authors provide more details on the proxy dataset used to train the budget allocator? Specifically, how is the proxy dataset selected, and how does it ensure adequate representation of non-IID distributions across diverse client tasks?
3. The paper references federated learning in related works, have you done any comparisons with FL methods that tackle non-IID distribution, especially regarding data and budget efficiency
4.  How sensitive is the budget allocator to differences across task types? Does the allocator need to be re-trained for different tasks?
5. Table 3 shows the result from Llama-2-7B but line 502 says Gemma-2B was used
6. Missing reference for ICL annotation selection under limited budget - "Mavromatis et al. Which examples to annotate for in-context learning? towards effective and efficient selection"

---

> ### Author Response · Authors · 2024-11-27
>
> __1. Task covered in  study is limited to classification tasks. It would benefit from proving  effectivenss across RAG and multihop reasoning.__
>
> Thank you for your thoughtful suggestion on inclusion of RAG & multihop reasoning tasks. Our main contribution in this paper is to demonstrate the feasibility of using ICL under non-IID conditions, and we focus on classification tasks as a starting point. While RAG & multihop reasoning are indeed more text-heavy and complex tasks, further experiments and exploration are needed to evaluate applicability of our method in those settings. This represents a promising direction for future work.
>
> We also note that many prior ICL-related studies [1][2][3] focus on classification tasks, which have proven to provide valuable insights to the research community. We believe our work aligns with this tradition while laying foundation for broader applications.
>
> [1] Lyu, Xinxi, et al. "Z-ICL: Zero-Shot In-Context Learning with Pseudo-Demonstrations." ACL. 2023.
>
> [2] Yoo, Kang Min, et al. "Ground-Truth Labels Matter: A Deeper Look into Input-Label Demonstrations." EMNLP. 2022.
>
> [3] Chen, Huiyao, et al. "Retrieval-style in-context learning for few-shot hierarchical text classification." TACL 2024.
>
> __2.  Paraphrasing based method to secure privacy during data retrieval - seems to have limited evaluation and more robust evaluation against other privacy-preserving techniques needs to be done to support  claim__
>
> Thank you for your insightful comment on evaluation of paraphrasing-based method. While paraphrasing is not main focus of our work, it serves to demonstrate that our framework can seamlessly integrate with plug-in privacy-preserving techs, which are orthogonal to current scope of our research. We chose paraphrasing as an example because prior studies [1][2][3] have successfully employed it for privacy preservation in LLM research.
>
> [1] ​​Zhang, Z., Zhang, J., Huang, J., Qu, L., Zhang, H., & Xu, Z. (2024). Fedpit: Towards privacy-preserving and few-shot federated instruction tuning. arXiv preprint arXiv:2403.06131.
>
> [2] Krishna, Kalpesh, et al. "Paraphrasing evades detectors of ai-generated text, but retrieval is an effective defense." NeurIPS (2024).
>
> [3] Yadav, V., Tang, Z., & Srinivasan, V. (2024, July). Pag-llm: Paraphrase and aggregate with large language models for minimizing intent classification errors. 47th ACM SIGIR.
>
> __3. Even when sharing similar task, clients might have different prompts in structure, length. Will this affect retrieval effectiveness since it heavily depends on  similarity between query and training examples?__
>
> Thanks for meaningful comment. We conducted additional experiments for Non-IID with feature-skew (same label distribution but different query distribution) to address this suggestions.
>
> We designed NonIID where 1 client contains only Yelp training samples, while another client contains only Amazon training samples. Yelp & Amazon share same label space (5-class classification), while they show different distributions on queries. On server, we use both Amazon&Yelp samples for test set, and perform t-SNE on test embedding with their budget values on each client. By this setting, we want to show our method intuition on ‘budget value’ and ‘sample embeddings’  still holds even it is feature skew with class-balance.
>
> As shown in [[client 1 (with only Yelp)]](https://anonymous.4open.science/r/Image-Materials-0C5F/mix-yelp-amazon-tsne-client0.png) and [[client 2 (with only Amazon)]](https://anonymous.4open.science/r/Image-Materials-0C5F/mix-yelp-amazon-tsne-client1.png),  clustering pattern is even more significant than previous class-based nonIID, indicating our claim still holds under text style non-IID. Also, we found Amazon test samples tend to assign all budget to Amazon client, while Yelp samples tend to assign all budget to Yelp client.
>
> To conclude, our method can also be applied to feature skew (style shifting) non-IID.
>
> __4.  Paper references federated learning in related works, comparison with FL methods in this setting?__
>
> Thank you for your insightful comment. Our current framework does not involve local training or global model aggregation, which differentiates it from FL setting. As such, widely-used FL methods like FedAvg and FedProx are not directly applicable to our approach, which is why they were not included in the experiments.
>
> __5. Typo on Llama-2-7B & line 502 Gemma-2B, and missing reference for ICL annotation under limited budget - "Mavromatis et al. Which examples to annotate for in-context learning? towards effective and efficient selection".__
>
> Thank you for pointing out these, we have corrected these in revised version (line 503 & related works).

---

> > ### Author Response · Authors · 2024-11-27
> >
> > __6. Relies on  assumption that a high quality proxy set on server. Although experiments show stability in data size, we want to see further experiments on how proxy quality or distribution affects  performance or alternative methods to reduce such dependency.__
> >
> > Thank you for the thoughtful comment on reliance on a high-quality proxy set. We conducted additional experiments to explore impact of proxy quality and distribution on performance under two different scenarios:
> >
> > 1. _same dataset but different label distribution._  The most simple case of “different distribution” can be different label distribution skew between proxy & test set. We conduct experiment on Subj with proxy only containing samples of one class. As shown in the table (last row), when label skew exists, performance of our method does decrease compared with using ideal proxy set (from $82.36\%$ to $70.17\%$). However, it is still higher than some baselines (zero-shot, singleton, uniform-budget and random-budget). Notice that "proxy-only" here uses a balanced proxy set for ICL inference, while our method with a single-class proxy set achieves similar performance (71.09% vs 70.17%). This indicates our method is not that bad even using extreme proxy set.
> >
> > |           | Subj  |
> > | ------| ----- |
> > | Zero-shot   | 50.55     |
> > | Proxy-only   | 71.09     |
> > | Singleton     | 50.00     |
> > | Social Learning  | 71.37     |
> > | Uniform-budget   | 63.20     |
> > | Random-budget  | 65.37     |
> > | $\infty$-budget    | 91.40     |
> > | Ours     | __82.36__ |
> > | Ours-proxy-label-skew | 70.17     |
> >
> > 2. _similar task but different dataset._ To evaluate a more extreme scenario, we used proxy sets from different datasets sharing the same task as the test set:
> >  - Amazon as proxy for Yelp Non-IID setting, evaluate on Yelp test
> >  - Yelp as proxy for Amazon Non-IID setting, evaluate on Amazon test
> >
> > Since Yelp & Amazon are both 5-class classification, we can consider this setting as using available dataset with similar task as test set to construct proxy. this setting demonstrates the use of available datasets for proxy construction. As shown in the table (last row),  for the Amazon setting with Yelp as the proxy, the performance drop is minimal, and our method still outperforms other baselines, except the ideal case. For the Yelp setting with Amazon as the proxy, our method even surpasses the ideal case.
> > |            | Amazon  | Yelp  |
> > | ------- | ---- | ---- |
> > | Zero-shot  | 24.70  | 31.23  |
> > | Proxy-only  | 28.43  | 31.85  |
> > | Singleton   | 24.03       | 29.44 |
> > | Social Learning   | 28.42   | 29.25|
> > | Uniform-budget    | 25.63  | 26.60  |
> > | Random-budget   | 25.69   | 27.72  |
> > | $\infty$-budget     | 32.70   | 34.80  |
> > | Ours       | __31.54__ | 35.48  |
> > | Ours-diff-proxy | 31.27  | __37.33__ |
> >
> > These results suggest that using open-source datasets with a similar task is a viable alternative when an exact match for the test distribution is unavailable.
> >
> > In conclusion, while having prior knowledge of the test set distribution is important, our experiments demonstrate that our method remains effective even with proxy sets differing in distribution, highlighting its robustness and practical applicability.
> >
> >
> > __7. More details on proxy set used to train  budget allocator? How is  proxy dataset selected, and how does it ensure representation of non-IID across diverse client tasks?__
> >
> > Thank you for insightful comment on selection & representativeness of proxy set. Ideally, proxy set should share same distribution as test set. In implementation, we randomly select samples from test set to form proxy set. It is important to note that proxy does not need to cover client’s local distribution but only needs to resemble test set distribution. As long as proxy samples capture clustering patterns of budget values (e.g., as seen in t-SNE), the performance on test set can be ensured.
> >
> > In practical scenarios, obtaining proxy set with exact same distribution as test set may be challenging. However, it is feasible to use datasets from other sources that share same task as test set. For instance, we demonstrated this by using Yelp as the proxy set for Amazon and vice versa. These experiments show that leveraging available datasets with similar tasks is a viable solution.
> >
> > In realistic setting, medical applicaiton for example, there are some available open-source data we can use as proxy set as long as they are same task as test set. For example, for Alzheimer’s disease detection using EHR, we can use [OHSU](https://www.ohsu.edu/alzheimers-disease-research-center/data-resources) [1] data as proxy; for metastatic cancer detection using EHR, we can use MIMIC-III [2].
> >
> > [1] Zhang, Xi Sheryl, et al. "Metapred: Meta-learning for clinical risk prediction with limited patient electronic health records." 25th ACM SIGKDD. 2019.
> >
> > [2] Johnson, Alistair EW, et al. "MIMIC-III, a freely accessible critical care database." Scientific data 3.1 (2016): 1-9.

---

> > > ### Author Response · Authors · 2024-11-27
> > >
> > > __8. How sensitive is budget allocator to differences across task types? Does allocator need to be re-trained for different tasks?__
> > >
> > > Thank you for your thoughtful question. The budget allocator learns the relationship between proxy sample embeddings and the query distributions of local datasets. Therefore, if local sample distributions change significantly (e.g., different task types), the allocator would need to be retrained.
> > > Here, we provide a simplified version of this “different task” setting: local clients use samples from one dataset (same as test set), while proxy set uses samples from a different dataset with a similar task. Again, we use experiment of "Yelp as Amazon's proxy" and and vice versa to demonstrate. This experiment shows if two tasks are similar, then _there is notable transferability in the budget allocator's effectiveness._.

---

> ### Author Response · Authors · 2024-12-01
>
> Dear reviewer, thank you for your thoughtful feedback and for providing a positive evaluation of our work. We greatly value your insights, which have been instrumental in improving our submission. As the discussion period is coming to a close, we wanted to kindly check if our rebuttal addressed your concerns satisfactorily. If there are any remaining questions or points you would like us to clarify, we would be happy to address them promptly.
>
> Thank you again for your time and effort in reviewing our work. We truly appreciate your contribution to this process.

---

> > ### Comment · Reviewer_GvJK · 2024-12-01
> >
> > I thank the author for their comprehensive responses, with additional experiments conducted to answer most of my questions. I am pleased to see those extra results and makes the story more complete. I will keep my scores since they are already positive.

---

> > > ### Author Response · Authors · 2024-12-01
> > >
> > > Thank you very much!

---

### Official Review · Reviewer_rqSc · 2024-10-31

**Soundness:** 2
**Presentation:** 3
**Contribution:** 2
**Rating:** 5
**Confidence:** 4

**Summary:**

This paper aims at addressing the issue of non-iid distributed  in-context examples (ICE) for in context learning. The authors
introduce an approach to tackle the distributed non-IID ICL problem by calculate the budget for different clients on the number of ICEs for different clients. The principle is that each client’s proper contribution (budget) should be designed according to the preference of each query for that client. This is done by the server who
will gather the optimal budget statistics using an existing proxy dataset on the server side. Basically, the idea is straightforward with limited novelty. The paper is well structured and the experiments are thorough. However, as mentioned, the novelty and technical depth is limited.

**Strengths:**

Pros:
1). The paper is well structured

2). The experiments are thorough.

**Weaknesses:**

Cons:

1). The novelty and technical depth is limited. The core idea is the server
will gather the optimal budget statistics using an existing proxy dataset on the server side. This however is pretty straightforward.

2). I think the work is highly related to the distributed RAG work. THe authors are suggested to include the discussion of the difference of existing distributed RAG works and compare with these approaches if possible.

3). The paper only uses small open-sourced LLMs, such as GPT-Neo-1.3B GPT-Neo-2.7B Llama-2-7B. Is that possible to provide results using larger ones, such 70B LLama-3.1 and other close-sourced ones, such as Claude 3.5 and GPT-4o?

**Questions:**

as shown in the weakness.

---

> ### Author Response · Authors · 2024-11-27
>
> __1.  novelty and technical depth is limited.  core idea is  server will gather  optimal budget statistics using an existing proxy dataset on  server side. This however is pretty straightforward.__
>
> To the best of our knowledge, there is no existing work that specifically investigates whether this solution is feasible, making our contribution novel and significant. Furthermore,  distributed non-IID setting we propose holds meaningful implications for research community, particularly due to its applicability in real-world scenarios, such as collaborations between medical institutions.
>
> We believe problem setting itself is critical, as it reflects practical challenges and opportunities that can inspire future research. Many influential works in  field, such as those on in-context learning (ICL) and chain-of-thought (CoT) reasoning, have demonstrated that even straightforward methodologies can lead to profound contributions when they address meaningful and impactful problems.
>
> Similarly, we emphasize importance of introducing a meaningful and relevant problem setting that aligns with real-world needs, which, in our view, is as valuable to  community as  sophistication of  proposed methodology.
>
>
> __2. I think  work is highly related to  distributed RAG work.  authors are suggested to include  discussion of  difference of existing distributed RAG works and compare with these approaches if possible.__
>
> Thank you for your valuable suggestion regarding  inclusion of distributed RAG-related works. We sincerely appreciate your insightful feedback and agree that discussing  differences between our approach and existing distributed RAG studies will help provide additional clarity and context to our contribution.
>
> In developing this work, we carefully considered related studies in distributed RAG. However,  challenges addressed by existing distributed RAG works differ from those tackled in our paper. For instance, [1] focuses on  creation of datasets for distributed RAG frameworks and explores LLM-based labeling techniques for engineering pipelines. Their research scope and methodology are distinct from ours and are not directly applicable to our specific problem setting. Similarly, [2] addresses resource consumption and real-time response challenges in distributed RAG, emphasizing local retrieval efficiency and answer accuracy. However, it does not account for  non-IID property in distributed settings. Additionally, [2] permits LLM deployment on partial local institutions, which is fundamentally different from our setting.
>
> Real-world distributed non-IID RAG scenarios present a more complex framework, involving numerous challenges that must be addressed for effective deployment. For example:
> - How can we effectively decompose a user query into subqueries while considering local knowledge distribution?
> What is  best way to assign these subqueries to different clients with varying local expertise?
>
> - How should we merge knowledge retrieved from multiple clients with overlapping expertise, and should we assign confidence levels to different clients for the same subqueries?
>
> - How can local retrieval process be accelerated when dealing with large local databases?
>
> These challenges represent broader avenues for exploration in distributed non-IID RAG. While our current work cannot directly compare with existing distributed RAG studies due to different settings, we believe it offers an interesting starting point for addressing such challenges. Specifically, our approach focuses on how to enable cooperation among clients with different knowledge distributions. By assigning preferences to clients based on their local knowledge distributions and employing an MLP to learn these distributions without transmitting complete local knowledge to a central server, we offer an intuitive method that could inspire future advancements in distributed non-IID RAG.
>
> We have included this discussion in the revised manuscript to further emphasize these distinctions and highlight unique aspects of our approach. Once again, thank you for your thoughtful feedback, which has been very helpful in refining our work.
>
> [1] Wang, Shuai, et al. "Feb4rag: Evaluating federated search in  context of retrieval augmented generation." Proceedings of  47th International ACM SIGIR. 2024.
>
> [2] Li, Jiaxing, et al. "EACO-RAG: Edge-Assisted and Collaborative RAG with Adaptive Knowledge Update." arXiv preprint arXiv:2410.20299 (2024).

---

> > ### Author Response · Authors · 2024-11-27
> >
> > __3.  paper only uses small open-sourced LLMs, such as GPT-Neo-1.3B GPT-Neo-2.7B Llama-2-7B. Is that possible to provide results using larger ones, such 70B LLama-3.1 and other close-sourced ones, such as Claude 3.5 and GPT-4o?__
> >
> > Thank you for the valuable suggestion. We conduct all baselines for Subj. As shown in following table, our method still outperforms other baselines using large scale model GPT-3.5. We have also added these results in Table 3 in  revised version.
> >
> > | Algorithm   | Zero-shot | Proxy-only | Singleton | Social Learning | Uniform-budget | Random-budget | \infty-budget | Ours   |
> > | ------------- | --------- | ---------- | --------- | --------------- | -------------- | ------------- | --------------- | --------- |
> > | gpt-3.5-turbo | 57.57   | 88.44   | 60.81   | 87.53      | 81.23     | 81.47     | 92.23      | __91.33__ |

---

### Official Review · Reviewer_HMZd · 2024-11-05

**Soundness:** 2
**Presentation:** 3
**Contribution:** 2
**Rating:** 6
**Confidence:** 4

**Summary:**

This paper addresses the challenge of distributed in-context learning under non-identically distributed (non-IID) data across multiple clients. It trains a budget allocator to dynamically allocates the number of in-context examples (ICE) retrieved from each client based on the relevance of query. Specifically, it trains a multi-layer perceptron (MLP) for each client to approximate the oracle budget, derived from a server-side proxy dataset, enabling efficient and targeted ICE retrieval per query. This paper additionally explores paraphrasing techniques to ensure data privacy in distributed contexts. Empirical results across multiple datasets show that this approach outperforms several baselines.

**Strengths:**

1. Distributed ICL under non-IID conditions is an interesting problem and aligns well with real-world scenarios. This paper explore the challenges under this setting, providing some meaningful insights.
2. The proposed method is simple yet effective across several benchmarks, with low training overhead.

**Weaknesses:**

1. In real-world scenarios, data distribution differences can manifest in multiple aspects, such as text length, style, etc., but this paper only focus on Non-IIDess at the class level. I strongly recommend the author to take more aspects into considerations.
2. Since the training of allocator does not require the label of examples, the experiments should not be limited to classfication tasks. The effectiveness of allocator on generation tasks remains to be validated.
3. The partition for non-IIDness in the main experiments is unreasonable. According to Table 7, for binary classification task like Subj and MR, each client only having access to data under one class is too extreme and can easily lead to biased predictions.

**Questions:**

The training of the allocator relies on a proxy dataset. In practice, the distribution of test data is usually unknown and obtaining a proxy dataset with the same distribution is unrealistic. What if the distribution of the proxy dataset differs from that of the test set?

---

> ### Author Response · Authors · 2024-11-27
>
> __1. Limited nonIID, can happen on text length, styles, etc. Authors only focus on class NonIID. Recommend for adding more nonIID setting.__
>
> Thank you for valuable suggestion on other nonIID. We appreciate it and agree that non-IID can manifest in various forms. In this paper, our primary goal is to demonstrate the feasibility of using ICL under nonIID conditions, focusing on class-level non-IID as a starting point. Given the scope of a single paper, it is challenging to cover all possible nonIID scenarios comprehensively.
>
> To address your suggestion and evaluate applicability of our method to other nonIID, we added additional experiment involving feature skew. We consider a scenario where class distributions are balanced across clients, but query embedding distributions differ (i.e., style shifting without class nonIID).
>
> In this setting: one client contains only Yelp training samples, while another client contains only Amazon training samples. Yelp & Amazon share same label space (5-class classification), while they show different distribution on queries. On server, we use both Amazon & Yelp samples as test set, and perform t-SNE on test sample's embedding with budget values on each client. By this setting, we want to show our method intuition on budget value and sample embeddings still holds for feature skew (style shifting).
>
>  As shown in [[client 1 (with only Yelp)]](https://anonymous.4open.science/r/Image-Materials-0C5F/mix-yelp-amazon-tsne-client0.png) and [[client 2 (with only Amazon)]](https://anonymous.4open.science/r/Image-Materials-0C5F/mix-yelp-amazon-tsne-client1.png), it is clear that clustering pattern is even more significant than previous class nonIID, indicating our claim still holds even without special class-based distribution. And it is very interesting to know that Amazon test samples will tend to assign all budget value to  Amazon client, while Yelp test samples will tend to assign all budget value to Yelp client.
>
> To conclude, we believe these results demonstrate that our method can be successfully applied to feature-skew (style-shifting) non-IID, further broadening its applicability. Thank you for your insightful suggestion, which has helped us expand scope of evaluation.
>
> __2. Since training of allocator does not require label of examples,  experiments should not be limited to classification tasks. Effectiveness on generation task need to validated.__
>
> Thank you for the thoughtful suggestion on inclusion of generation tasks. We appreciate it and agree that exploring such tasks could be an interesting direction for future work. However, primary focus of this paper is to demonstrate the feasibility of using ICL under non-IID conditions, with text classification as a representative task.
>
> Many prior ICL-related works [1][2][3] have similarly concentrated on classification tasks, providing significant value to research community. Our work builds on this foundation by examining nonIID for classification in detail, which we believe is a substantial and valuable contribution.
>
> Furthermore, generation tasks in distributed nonIID pose unique challenges and require significant exploration of task-specific nonIID designs & experimental setups. Addressing these would broaden scope of our current work beyond its intended focus. For this reason, we believe it is more appropriate to limit scope of this paper to classification tasks.
>
> We hope this clarifies our rationale, and we appreciate your valuable feedback, which has provided us with useful ideas for extending this work in future research.
>
> [1] Lyu, Xinxi, et al. "Z-ICL: Zero-Shot In-Context Learning with Pseudo-Demonstrations." ACL. 2023.
>
> [2] Yoo, Kang Min, et al. "Ground-Truth Labels Matter: A Deeper Look into Input-Label Demonstrations." EMNLP. 2022.
>
> [3] Chen, Huiyao, et al. "Retrieval-style in-context learning for few-shot hierarchical text classification." TACL.2024.
>
> __3.  partition for nonIIDness in main experiments is unreasonable. According to Table 7, for binary classification like Subj & MR, each client only having one class data is too extreme and can easily lead to biased predictions.__
>
> Thank you for suggestion on experiment design for binary classification. We added experiment on NonIID based on Dirichlet, where each client has samples from both classes with class imbalance. For detailed distribution on each client, please check [[MR]](https://anonymous.4open.science/r/Image-Materials-0C5F/mr-dist.png) & [[Subj]](https://anonymous.4open.science/r/Image-Materials-0C5F/subj-dist.png). As shown in table, our method still outperforms others under non-extreme nonIID for MR&Subj.
>
> |        | MR  | Subj  |
> | ----- | ----- | ------ |
> | Zero-shot    | 73.95   | 50.55   |
> | Proxy-only   | 70.40  | 71.09  |
> | Singleton    | 64.16   | 73.80 |
> | Social Learning   | 58.85  | 76.95 |
> | Uniform-budget   | 52.85  | 77.80  |
> | Random-budget  | 53.50  | 77.85  |
> | $\infty$-buddget  | 77.20  | 91.40  |
> | __Ours__   | __75.53__ | __82.80__  |

---

> > ### Author Response · Authors · 2024-11-27
> >
> > __4.  Training of allocator relies on a proxy dataset. In practice, the distribution of test data is usually unknown & obtaining a proxy dataset with same distribution is unrealistic. What if distribution of proxy set differs from that of test set?__
> >
> > Thank you for raising this important point about potential distribution differences between proxy & test set. We agree that controlling distribution shifts is critical and have conducted additional experiments to address this concern under two different scenarios:
> > 1. _same dataset but different label distribution._  Most simple case of “different distribution” can come from different label distribution skew between proxy & test set. We conduct experiment on Subj with proxy set only containing samples of one class. As shown in the table (last row), while performance does decrease compared to the ideal proxy set (from 82.36% to 70.17%), our method still outperforms several baselines such as zero-shot, singleton, uniform-budget, and random-budget. Notice that "proxy-only" here uses a balanced proxy set for ICL inference, while our method with a single-class proxy set achieves similar performance (71.09% vs 70.17%). This indicates our method is not that bad even using extreme proxy set.
> >
> > |                                      | Subj       |
> > | ------------------------------ | ----------- |
> > | Zero-shot                      | 50.55     |
> > | Proxy-only                    | 71.09     |
> > | Singleton                      | 50.00     |
> > | Social Learning            | 71.37     |
> > | Uniform-budget            | 63.20     |
> > | Random-budget           | 65.37     |
> > | $\infty$-budget             | 91.40     |
> > | Ours                             | __82.36__ |
> > | Ours-proxy-label-skew | 70.17     |
> >
> >
> >
> > 2. _similar task but different dataset._ To evaluate a more extreme case, we used proxy sets from different datasets that share the same task as the test set. Specifically:
> > - Amazon as proxy for Yelp Non-IID setting (evaluate on Yelp test)
> >
> >  - Yelp as proxy for Amazon Non-IID setting (evaluate on Amazon test)
> >
> >  Since Yelp & Amazon share similar task, this setting simulates using available datasets for proxy construction. As shown in the table (last row), the results indicate that:
> > - for Amazon setting use Yelp as proxy,  performance drop of our method is slight, and our method still outperforms other baselines, except the ideal case.
> >
> > - for Yelp setting using Amazon as proxy, our method unexpectedly achieves even better performance than the ideal case.
> >
> > |                             | Amazon    | Yelp      |
> > | ---------------------- | -------------- | ----------- |
> > | Zero-shot             | 24.70        | 31.23     |
> > | Proxy-only            | 28.43       | 31.85     |
> > | Singleton              | 24.03       | 29.44     |
> > | Social Learning    | 28.42       | 29.25     |
> > | Uniform-budget    | 25.63       | 26.60     |
> > | Random-budget   | 25.69       | 27.72     |
> > | $\infty$-budget     | 32.70       | 34.80     |
> > | Ours                     | __31.54__ | 35.48     |
> > | Ours-diff-proxy     | 31.27       | __37.33__ |
> >
> > These results suggest that, even when an exact match for the test distribution is unavailable, it is feasible to use open-source datasets with a similar task to construct proxy set for our method.
> >
> > In conclusion, while having prior knowledge of test set distribution is valuable, our experiments demonstrate that using a proxy set with similar task is a practical and effective solution.

---

> > > ### Comment · Reviewer_HMZd · 2024-11-28
> > >
> > > Thanks for the response. It has addressed most of my concerns. I have decided to raise my score. I find it interesting that using Amazon as a proxy for the Yelp Non-IID setting leads to higher accuracy. I recommend that the authors explore this phenomenon in more depth and provide further discussion on it.

---

> > > > ### Author Response · Authors · 2024-12-01
> > > >
> > > > Thank you for your kind words and appreciation of our work. We are grateful for your interest in our findings and agree that it is indeed intriguing that using Amazon as the proxy set for the Yelp Non-IID setting leads to higher accuracy. Considering that this configuration outperforms both the ideal case of our method and the $\infty$-budget method, we offer the following analysis and hypotheses to explain this phenomenon:
> > > > - _Query embedding distribution relationship:_ The relationship between the query embedding distributions of Amazon and Yelp may play a key role. When we visualize [t-SNE for samples from both Amazon & Yelp](https://anonymous.4open.science/r/Image-Materials-0C5F/mix-yelp-amazon-tsne-client0-color-source.png), it becomes evident that Amazon and Yelp samples do not uniformly overlap. Instead, most Amazon samples are located on a distinct side of the cluster formed by Yelp samples. This suggests that the relevance scores derived from Amazon samples are influenced by "single-sided" perspectives of the Yelp cluster, rather than by intra-cluster samples, potentially altering the measurement of "relevance."
> > > >
> > > > - _Introduction of diversity in retrieved samples:_ Retrieved samples contribute not only relevance information but also other factors such as diversity. A proxy set that introduces a balance of relevance and diversity may help construct an in-context example set that improves the performance of the final inference result. This might explain the enhanced accuracy observed in this configuration.
> > > >
> > > > Thank you again for highlighting this fascinating phenomenon and for your valuable suggestion to explore it further. We will do our best to include additional experimental results and a more detailed discussion in the camera-ready version, if possible.

---

### Official Review · Reviewer_aykb · 2024-11-12

**Soundness:** 2
**Presentation:** 2
**Contribution:** 3
**Rating:** 5
**Confidence:** 4

**Summary:**

This paper tackles the challenge of distributed non-IID in-context learning (ICL) for LLMs, where data is spread across clients with differing distributions. The paper first shows that uniform in-context examples would fail on non-IID situations. Then the authors propose a method to optimize the allocation of a limited in-context examples (ICEs) budget by training a budget allocator. This allocator predicts query-specific budgets for each client, addressing the inefficiencies of uniform allocation under non-IID settings. The method outperforms baseline approaches like random and uniform budgets in experiments, improving ICL performance on distributed datasets.

**Strengths:**

1. This paper highlights the important yet underexplored problem of distributed non-IID in-context learning (ICL), offering fresh insights into a new problem.
2. The paper presents some valuable experimental results demonstrating the poor performance of distributed non-IID ICL under simple uniform budget allocation, effectively validating the significance of the problem.
3. The method proposed in this paper have improved the performance in a simple while effective way.

**Weaknesses:**

1. The paper does not provide concrete examples of distributed non-IID ICL scenarios. So i don't get that given the server can request budgeted samples from each client, why can't these samples be used to simulate a comprehensive, unbiased retrieval pool for inference?

2. The training dataset for the allocator is constructed by retrieving k samples per query from each client, combining these 𝐶×𝑘
samples to simulate a unified dataset. However, this simulation raises questions. If the simulation is reasonable (e.g., with large 𝑘), why not directly perform retrieval from this simulated dataset instead of training an allocator? If the simulation is unreasonable, can this dataset still be valid for training the allocator?

3. In Figure 8, the proxy size appears to have minimal influence, which is surprising. Since the allocator is trained using questions from the proxy dataset, a poor match between the proxy and test set questions should bias the collected data and hinder the training of a good allocator. However, this surprising result lacks a detailed explanation.

4. The conclusion in Section 3-Observations that query embeddings can determine budget assignments is based on observed clustering patterns in oracle budgets corresponding to different queries. This conclusion may be too strong, as other factors, such as the specific distribution of client data, might play a criticle role. For instance, in the experiments, non-IID clients are constructed based on classes, and these class-based distributions likely influence budget assignments significantly.

5. The experiments simulate non-IID clients based on data classes. Could an LLM directly infer which classes are relevant for a query and decide sample allocations accordingly? Since the allocator effectively behaves like a classifier for assigning budgets based on query classes, a straightforward rule-based budget allocation using known client classes might perform comparably.

6. There are multiple spelling mistakes in the paper. For instance, in Section 2.2, the first two sentences describing the pipeline use k_c with seemingly different meanings, leading to confusion.

**Questions:**

1. Why not evaluate using inherently distributed non-IID datasets? Is there an available one or not?
2. How are proxy datasets constructed, and do they ensure coverage of all clients? The description suggests the proxy set is sampled directly from the test set, but what real-world scenario does this correspond to, and how would such a proxy dataset be realistically constructed? Can you provide specific examples of a proxy dataset in real-world application?

---

> ### Author Response · Authors · 2024-11-27
>
> __1. Concrete examples of distributed non-IID ICL scenarios. Why can't these samples be used to simulate a comprehensive, unbiased retrieval pool for inference?__
> - Sorry for the confusion. We have added detailed examples to Appendix G of revised version to address your concerns and assist future readers in understanding this research. Please check it (due to character limitation, we do not post it here).
>
>  - __“Why not use requested budgeted samples from each client to construct a better retrieval pool for inference?”__ Yes, it is a meaningful question during our framework design. We provide  following reasons to argue that this solution is impractical in real-word settings.
>    - During  budget allocator training stage, each client only needs to upload  “relevant scores” of  local top relevant samples, rather than  raw samples consisting of input query $x$ and  corresponding label $y$, that is,  server does not have  raw information of each training sample from  client. Thus, this information collected for budget allocator training cannot be used as an ICL retrieval pool.
>
>     - During  inference stage (with trained budget allocator), can we accumulate  collected local samples of  previous test queries and use these as a later retrieval pool? Due to privacy concerns, server platform can use local samples to perform inference while not allow caching these samples.
>
> To conclude, a “comprehensive retrieval pool based on budgeted samples” is not applicable in our distributed non-IID scenario due to data pricing and privacy concerns.
>
> __2. If simulation of retrieval for proxy is reasonable (e.g., with large 𝑘), why not directly perform retrieval from this simulated data instead of training an allocator? If simulation is unreasonable, can this dataset still be valid for training  allocator?__
>
> During construction of training data for allocator, we do not request local clients to send raw local training samples (that means, no label information) to server. Local clients only send “relevant scores” of local samples to server, considering cost of both communication and data pricing. Then we use collected ‘relevant scores’ to estimate “oracle budget” for each query in proxy dataset. That means, without these estimated “oracle budget” for queries in proxy set, it is impossible to train budget allocator.
> - _If simulation is reasonable (with large k):_ It is impossible for server to do ICL based on this collected training dataset, since retrieval process in budget allocator training stage only contains similarity between samples’ queries while no sample labels collected during this stage.
>
> - _If simulation is unreasonable:_ If it is allowed for transformation of relevance scores on only small k local samples (rather than a large k), then it can happen that estimated “oracle budget” is not accurate enough, which can lead to a sub-optimal budget allocator. Consequently,  budget prediction during inference stage would result in sub-optimal budget allocation and lead to suboptimal ICL performance. An extreme case is, given 4 clients and server-side ICE budget for each query is 16, each client is only allowed to send  top-1 relevant score to server. Then  server can only estimate each local “oracle budget” as [1,1,1,1] (since overall number is even less than 16), which provides no useful information for training on budget allocator.
>
> __3. Detailed explanation on Figure 8, why show robustness on different size of proxy set?__
>
> -  We add experiment results on extremely small proxy set, that is proxy set with only 100 and 50 samples. As shown in [[proxy robustness results]](https://anonymous.4open.science/r/Image-Materials-0C5F/rebuttal-proxy-size-robustness.png), with extremely small proxy set, performance does drop a lot (from 80+% to lower than 75%, even lower than 65%).  results we present in our paper only shows  results of proxy size from 300 to 700. Sorry for  confusion.
> - Also, as shown in [[proxy robustness results]](https://anonymous.4open.science/r/Image-Materials-0C5F/rebuttal-proxy-size-robustness.png), we separately draw performance curve on different quantization resolution $\delta$. As shown in figure, given a fixed quantization resolution value (for $\delta=3$, $\delta=4$), performance slightly increases when proxy size changes from 300 to 700.  figure 8 presented in paper is  “best result” for each proxy size (red curve). Thus it shows limited change. This may cause confusion, as mentioned in your comment.
>
> We also present complete result figure in revised version. Thank you for pointing out this.
>
> __4. Why no inherently distributed non-IID datasets? Is it available?__ Thank you for  suggestion. As far as we know, this is  first paper on ICL tasks under distributed non-IID settings. Therefore, we didn't find any available open-source inherently distributed non-IID datasets.

---

> ### Author Response · Authors · 2024-11-27
>
> __5. Conclusion in section 3-observation on query embedding and budget assignment is too strong. This may be impacted by special non-IID in experiment. Intuition may not work under class-balanced setting.__
>
> Thank you for pointing out this. We claim that main reason our method works is not because of special class-based Non-IID setting. To show this, we add extra NonIID setting with feature skew but class-balance.
> We designed NonIID setting that 1 client contains only Yelp training samples, while another client contains only Amazon training samples. Yelp & Amazon share same label space and similar task, while they show different distribution on queries. On server, we use both Amazon&Yelp samples for test, and perform t-SNE on test embedding and their budget values on each client. By this setting, we want to show our claim on budget value and sample embeddings still holds even under feature skew with class-balance.
>  As shown in [[client 1 (with only Yelp)]](https://anonymous.4open.science/r/Image-Materials-0C5F/mix-yelp-amazon-tsne-client0.png) and [[client 2 (with only Amazon)]](https://anonymous.4open.science/r/Image-Materials-0C5F/mix-yelp-amazon-tsne-client1.png), clustering pattern is even more significant than previous class-based non-IID, indicating our claim still holds even without special class-based distribution. Further, we emphasize that our method design has no relation to special label distribution.
>
> __6. Proposed allocator  behaves like classifier for assigning budgets based on query classes, a straightforward rule-based budget allocation using known client classes might perform comparably.__
>
> Thank you for raising this important point about the behavior of our proposed allocator. While it may seem that our method functions like a straightforward rule-based allocator, we would clarify that our method is not simply trying to learn clients' local class distribution. Instead, our method tries to learn clients' local _query embedding distribution_. We use experiment where 1 client with only Yelp samples and 1 client with only Amazon samples to explain. In this setting, two clients share same local class distribution, which is balanced over 5 classes. If our method only relies on local class distribution, then each test query should assign equal budgets to two clients. However, as shown in [client 1 (with only Yelp)](https://anonymous.4open.science/r/Image-Materials-0C5F/mix-yelp-amazon-tsne-client0.png) and [client 2 (with only Amazon)](https://anonymous.4open.science/r/Image-Materials-0C5F/mix-yelp-amazon-tsne-client1.png), most queries assign all budget to client with most similar query distribution (from same dataset), while assigning 0 budget to client with different query distribution (from another dataset). This indicates that our method does not simply learn local class distribution, but learns query embedding distribution, which can be a more complex problem.

---

> > ### Author Response · Authors · 2024-11-27
> >
> > __7. Does construction of proxy set ensure coverage of all clients? How would such a proxy dataset be realistically constructed?__
> >
> > Thank you for raising this thoughtful question about the construction and applicability of the proxy set. We would like to clarify that primary purpose of proxy set is to approximate distribution of test set, rather than to achieve coverage of clients. In real-world applications like medical area, we can use available open-source dataset to construct proxy, as long as they have similar task with real test set. For example, for Alzheimer’s disease detection using EHR, we can use [OHSU](https://www.ohsu.edu/alzheimers-disease-research-center/data-resources) [1] dataset; for metastatic cancer detection using EHR, we can use MIMIC-III [2].
> >
> > To verify our method under setting where proxy is constructed using other dataset sharing similar task with test set, we conduct following experiment:
> > - Amazon as proxy for Yelp Non-IID setting, evaluate on Yelp test
> > - Yelp as proxy for Amazon Non-IID setting, evaluate on Amazon test
> >
> > Since Yelp&Amazon share similar task, we consider this as using available open-source data as proxy. As shown in  table ( last line shows  performance of this setting), for  Amazon setting using Yelp as proxy, performance drop of our method is slight, and it still outperforms other baselines, except  ideal case. While for Yelp setting using Amazon as proxy, our method shows even better performance thanideal case. Thus, we think it is feasible to use open-source data with similar task to construct proxy set, and  performance of our method is still acceptable, which shows applicability in real-world.
> >
> > |                             | Amazon    | Yelp      |
> > | ---------------------- | -------------- | ----------- |
> > | Zero-shot             | 24.70        | 31.23     |
> > | Proxy-only            | 28.43       | 31.85     |
> > | Singleton              | 24.03       | 29.44     |
> > | Social Learning    | 28.42       | 29.25     |
> > | Uniform-budget    | 25.63       | 26.60     |
> > | Random-budget   | 25.69       | 27.72     |
> > | $\infty$-budget     | 32.70       | 34.80     |
> > | Ours                     | __31.54__ | 35.48     |
> > | Ours-diff-proxy     | 31.27       | __37.33__ |
> >
> >
> > [1] Zhang, Xi Sheryl, et al. "Metapred: Meta-learning for clinical risk prediction with limited patient electronic health records." 25th ACM SIGKDD.
> >
> > [2] Johnson, Alistair EW, et al. "MIMIC-III, a freely accessible critical care database." Scientific data 2019.

---

> ### Author Response · Authors · 2024-12-01
>
> Dear reviewer aykb, thank you for taking the time to provide detailed suggestions on our submission. We have carefully responsed each of your comments in our rebuttal, and we truly appreciate the opportunity to clarify and expand upon our work based on your valuable insights.
>
> As the discussion period is nearing its end, we wanted to kindly follow up to see if our responses addressed your concerns satisfactorily. If there are any remaining points or additional questions, we would be happy to provide further clarification.
>
> Thank you again for your time and effort in reviewing our work. Your feedback is invaluable, and we greatly appreciate your engagement with our submission.

---

### Meta-Review · Area_Chair_43Ej · 2024-12-16

**Metareview:**

In this paper, the authors proposed a new setting for ICL, where training data is stored in a distributed manner.

There are some major concerns raised by the reviewer. 1, The assumption of the proposed method may not be practical. Though the authors added some experiments, the assumption is still not convincing. 2, Though the problem setup looks new, the novelty of the proposed method is technically limited. 3, The study of the proposed new problem setup is only restricted to classification tasks. It is not a convincing reason to argue that previous ICL studies were only focused on classification problems in the authors' rebuttal.

By considering the concerns mentioned above, this paper does not meet the acceptance standard for ICLR.

**Additional Comments On Reviewer Discussion:**

The reviewers still have concerns about the assumption, applications and experiments of the proposed problem setup and method.

---

### Decision · Program_Chairs · 2025-01-22

Reject